# Quantifying archaeo-organic degradation – A multiproxy approach to understand the accelerated deterioration of the ancient organic cultural heritage at the Swedish Mesolithic site Ageröd

**Adam Boethius**[1]*, **Hege Hollund**[2], **Johan Linderholm**[3], **Santeri Vanhanen**[4], **Mathilda Kjällquist**[4], **Ola Magnell**[4], **Jan Apel**[5]

1 Department of Archaeology and Ancient History, Lund University, Lund, Sweden, 2 Museum of Archaeology, University of Stavanger, Stavanger, Norway, 3 Department of Historical, Philosophical and Religious Studies, Umeå University, Umeå, Sweden, 4 The Archaeologists, National Historical Museums, Lund, Sweden, 5 Department of Archaeology and Classical Studies, Stockholm University, Stockholm, Sweden

* adam.boethius@ark.lu.se

## Abstract

Despite a growing body of evidence concerning accelerated organic degradation at archaeological sites, there have been few follow-up investigations to examine the status of the remaining archaeological materials in the ground. To address the question of archaeo-organic preservation, we revisited the Swedish, Mesolithic key-site Ageröd and could show that the bone material had been subjected to an accelerated deterioration during the last 75 years, which had destroyed the bones in the areas where they had previously been best preserved. To understand why this has happened and to quantify and qualify the extent of the organic degradation, we here analyse the soil chemistry, bone histology, collagen preservation and palaeobotany at the site. Our results show that the soil at Ageröd is losing, or has already lost, its preservative and buffering qualities, and that pH-values in the still wet areas of the site have dropped to levels where no bone preservation is possible. Our results suggest that this acidification process is enhanced by the release of sulphuric acid as pyrite in the bones oxidizes. While we are still able to find well-preserved palaeobotanical remains, they are also starting to corrode through re-introduced oxygen into the archaeological layers. While some areas of the site have been more protected through redeposited soil on top of the archaeological layers, all areas of Ageröd are rapidly deteriorating. Lastly, while it is still possible to perform molecular analyses on the best-preserved bones from the most protected areas, this opportunity will likely be lost within a few decades. In conclusion, we find that if we, as a society, wish to keep this valuable climatic, environmental and cultural archive, both at Ageröd and elsewhere, the time to act is now and if we wait we will soon be in a situation where this record will be irretrievably lost forever.

**Data Availability Statement:** All relevant data are within the paper and its Supporting Information files.

**Funding:** We are grateful for the financial support from: JA, Crafoord foundation, nr. 20180631 https://www.crafoord.se/. AB, Lennart J. Hägglunds foundation for archaeological research and education http://hagglundsstiftelse.se/. AB, the Swedish National Heritage Board, RAÄ-2018-3237, https://www.raa.se/. The funders had no role in study design, data collection and analysis, decision to publish, or preparation of the manuscript.

**Competing interests:** The authors have declared that no competing interests exist.

## Introduction

When working with organic remains from archaeological sites, researchers have noticed that remains stored at museums are often in a better condition compared to what is recovered on more recent excavations. In an attempt to validate the accuracy of this tacit yet commonly held opinion we conducted a case study at the famous Swedish Middle-Mesolithic site Ageröd (section I:HC). The site was deemed appropriate for this purpose due to the large amounts of organic remains recovered on two former excavations of the site (in the 1940s and again in the 1970s) and because it is located in a secluded part of southern Sweden, which has not seen any major road constructions, railroads or modern buildings in the close vicinity of the site. The intrusions to the site do not, in general, exceed the minimum damage done to most other archaeological wetland sites in Northern Europe; that is, the site has been drained with low technological means (hand-dug narrow drainage ditches) and no mechanical pumps or major drainage channels have been used. Thus, the background pollution (acidic precipitation and exhaust etc.), climate change (leading to larger fluctuations in groundwater levels due to warmer summers), or previous archaeological excavations at the site have not impacted the local area around Ageröd more than at most other archaeological wetland sites in Northern Europe.

The 2019 re-investigation of Ageröd demonstrated that the bone remains at the site are threatened by accelerating destruction and that the previously best-preserved areas have now become the worst areas for bone preservation [1]. The excavation and zooarchaeological analysis of the osseous remains recovered on all three excavation campaigns (a total of 4240 bone fragments from Ageröd I were determined to family or species level and used in the study [1]) highlighted the problems of bone deterioration and showed that the osseous remains are suffering badly from accelerated degradation and revealed that in some areas this has destroyed the 9000-year-old bones, which only 75 years ago were well preserved. In an attempt to investigate the archaeo-organic preservation conditions, quantify the ongoing degradation and understand why the bones are rapidly deteriorating, a multiproxy approach to investigate different aspects of organic preservation and the soil properties related to the organic remains recovered at the 2019 re-excavation at the site is used. By investigating the soil chemical properties and by relating them to bone histological analyses, collagen preservation and the palaeobotany at the site, questions of how organic preservation has changed during the last seven decades and what might have caused the changes are answered and discussed. The present study shall be viewed as a part of investigating the prerequisites for the future survival of our long-term archaeo-environmental archive of climatic and environmental changes and/or its relation to past human cultural interaction [2–9], in a time when reports of ongoing and accelerated destruction of this valuable record emerge from all over the world [1, 10–22].

### Site description

Ageröd I is located on the edges of a peat bog to the north of Lake Ringsjön in central Scania, southern Sweden, which at the time of occupation was part of a shallow lake system connected to what is today Lake Ringsjön (Fig 1). The site was occupied in the Middle Mesolithic period, around 8700–8200 cal BP.

Ageröd I was found in the 1930s [23] and first excavated in 1946–1949 [24]. The site was then revisited in 1972–1974 [25]. On both these excavation campaigns, large numbers of organic finds were made, and the material from both excavations is stored at the Historical Museum at Lund University (LUHM). During the 1940s excavations, the site was divided into five major sections: A, B, C, HC, and D (see Fig 3 in [26]). Ageröd I consists of three

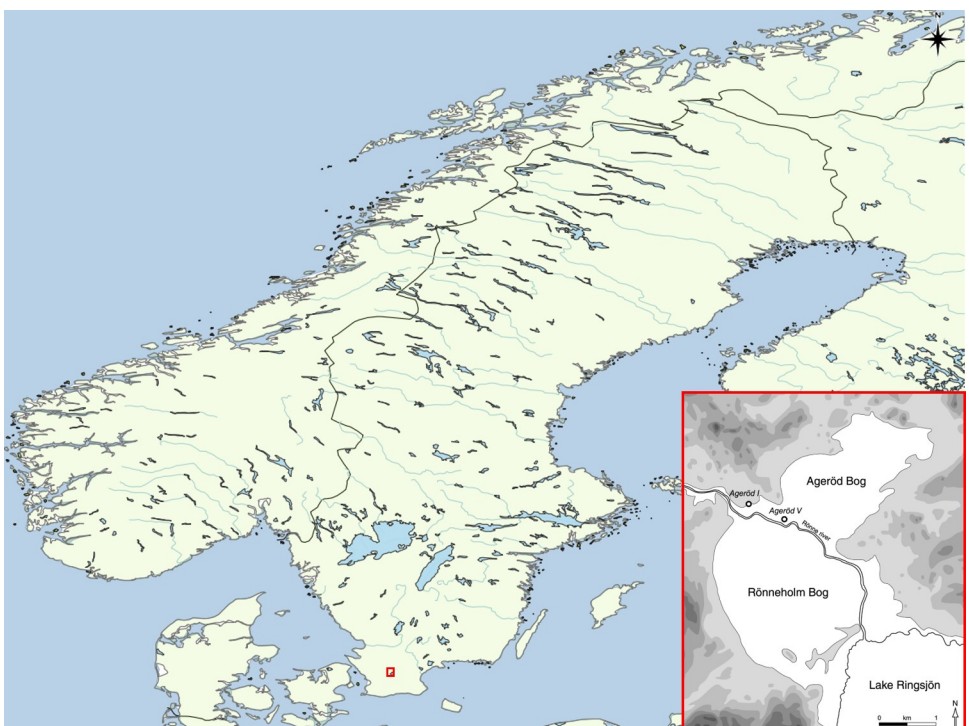

**Fig 1. Map of Scandinavia zoomed in on the area of the Ageröd I site located on the ancient shore of the former shallow lake.** World map generated with QGIS 3.10 using the Natural Earth data set. Lower right drawing of the Ageröd area by Arne Sjöström. Image freely available through CC by 4.0 licence by PLOS ONE and previously published in [1].

settlements where section A, C and HC intersect one settlement and Ageröd I:B and I:D intersects two others [25]. For more detailed information about the site see [1].

## Material and methods

### Permissions

The 2019 excavation of Ageröd was conducted with permission from the County Administrative Board, Scania, Sweden (reference number 431-3998-2019), in accordance with Swedish legislation. The permit allowed excavation with minor intrusions to meet the specified aims of investigating the preservation status at the site, and it allowed both destructive and non-destructive follow up analyses on the recovered remains to generate data on organic degradation. The archaeological remains recovered on the excavation are temporarily stored at The Archaeologists, National Historical Museums, Lund, Sweden, but will be transferred to LUHM, Lund, Sweden, where they will be permanently deposited along with the remains from previous excavation campaigns at Ageröd.

Comparative analyses were done on bone remains recovered on the 1940s and 1970s excavation of Ageröd [1]. The remains from these excavations are stored at LUHM under the deposition numbers LUHM 28977 and LUHM 80910 and are available at the museum upon request. An overview of all analysed specimens is presented in the Material section below. The permission obtained from LUHM included the sampling of six bones to conduct histological analyses (ca 3 g of bone per sample) and an additional 12 bone samples (0.5–1 g each) to investigate collagen deterioration (decisions 08-05-2019 and 29-08-2019).

### The excavation

In May 2019, we returned to Ageröd I. Targeted re-excavation was facilitated by using rectified GIS-data, obtained by relating aerial photographs from the 1950s, where the original trenches could still be seen, to the original site drawings and plans [24–26]. The positioning was tested for accuracy in the field by locating the fix-points that had been hammered and drilled into large stones during previous excavations at the site. Five 1x1 meter trenches were excavated and water sieved by hand. The trenches were strategically placed close to areas where organic remains had previously been abundant and the recovery of new organic remains and soil samples facilitated comparison with previously extracted organic materials. The trenches were also dispersed to different parts of the site corresponding to varying degrees of wetness, from the driest area in zone 1 to the wettest area in zone 3 (Fig 2). No trenches were made in zone 4,

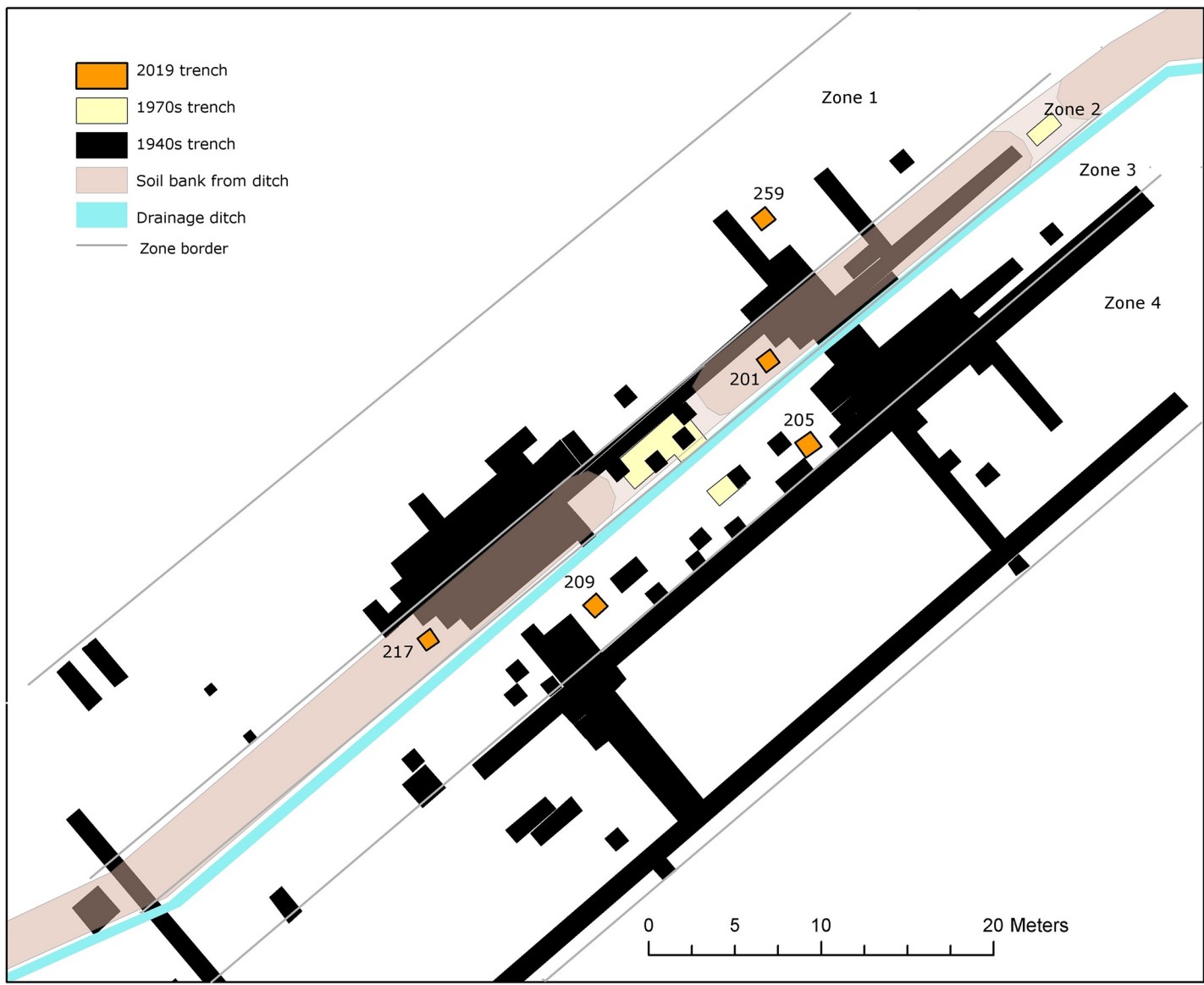

**Fig 2. The 1x1 meter trenches from 2019 in relation to the former excavation trenches, the ditch draining the bog and the soil bank of excavated ditch material from its establishment and maintenance.** The zone divisions were set to enable the study of differences in bone preservation between the driest (zone 1) and the wettest (zone 4) conditions. Image freely available through CC by 4.0 licence by PLOS ONE and previously published in [1].

which is located further out in deeper parts of the peat bog. For detailed information about the excavation and zooarchaeological analyses of the bone remains from all excavation campaigns of Ageröd I see [1].

## Soil chemistry

A total of 13 soil samples from the 2019 excavation were selected and sent to the Environmental Archaeology laboratory at Umeå University for analyses (see Results section Tables 3 and 4). The samples were unequally distributed in the five trenches, with one soil sample from the two trenches lacking bone remains, 209 and 259, four soil samples from trenches 201 and 205 and three from 217 (cf. Fig 3 for a stratigraphic overview of the location of the soil samples, trench 259 is not shown because its stratigraphy was disturbed by a previous and undocumented trench). No soil analyses were done on the re-deposited soil in the soil bank from trenches 201 and 217 (see [1] for further discussion of the different layers).

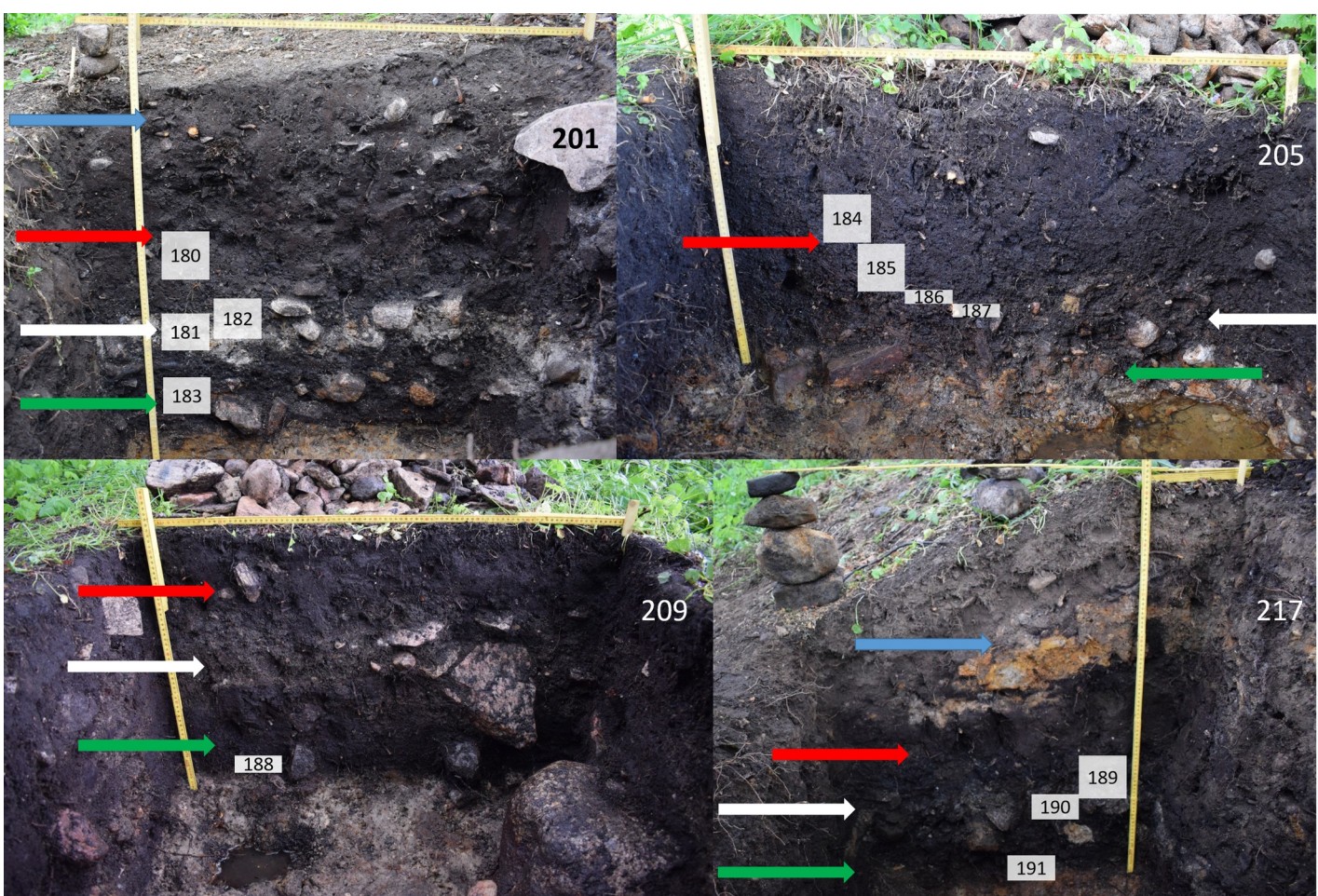

**Fig 3. Sections from the four undisturbed trenches excavated in 2019.** Blue arrow shows added soil from the drainage ditch, red arrow shows the upper peat layer, white arrow shows the white (archaeological) cultural layer and green arrow shows the lower peat layer. The different layers are most evident in trench 201, in trench 205 the lower peat layer is almost completely gone and difficult to detect as it has deteriorated and become merged with the white cultural layer. Trench 217 has the largest amounts of added soil from the drainage ditch on top of the originally deposited layers, and the ditch has been dug down into the moraine here as shown with the orange morainic soil in the added soil layer. The white numbered squares indicate the depth of the different soil samples (see Table 3 for depth information). The used terminology of the stratigraphic units/layers follow the original from the 1940´s. Image modified from original created for [1] and is freely available from PLOS ONE through CC by 4.0 licence.

Stratigraphic soil sampling was conducted at various points in the different trenches. Several analytical techniques were applied to the samples, including X-ray fluorescence spectroscopy (XRF), phosphate fractionation, loss on ignition, pH and magnetic susceptibility.

Before the analyses were made, all samples were dried at 30˚C. The samples were then passed through a 1.25 mm sieve and any presence of material of cultural significance noted (such as bone, charred material, flint etc.). The soil chemical analyses employed here follow the methodological approach of [27, 28] for phosphate fractionation. The use of various approaches to phosphate analysis has been discussed elsewhere [29, 30], and the pros and cons of using weak acid extraction as opposed to total dissolution techniques have been a matter of debate in geoarchaeology. There is a certain method response depending on the soil-sediment system at hand. In this case, the Citric soluble phosphate is used as an indicator of the intensity build-up of archaeological layers and to study the movement of P and Fe in the groundwater [31]. The Fe system is proxy-analysed by using magnetic susceptibility [32]. For element analysis, XRF was applied and a multitude of studies have been conducted in the field of Geoarchaeology [33–35]. The parameters analysed and abbreviations used are explained in Table 1.

These methods have been developed and adapted for soil prospection and the bulk analysis of occupation soils and features and provide information on various aspects concerning phosphate, iron, redox potential and other magnetic components and total organic matter in soils and sediments, and their relationship to phosphate [28, 38, 41, 42].

## Bone histology

Eight bone fragments from three different excavation units (201, 205 & 217) and different stratigraphic depths were sampled for histological analyses (see Results section Table 5), also representing different types of species and bone elements. In addition, six samples of material from the previous excavation campaigns, in the 1940s and the 1970s, were collected. Taphonomic analyses at a macroscopic level have revealed the extent of surface weathering from all three excavation campaigns [1], including the bones sampled for histological analyses.

Fragments of ca 2x1 cm were cut, embedded in resin and polished to produce thin-sections of 30–50 microns for study using transmitted normal and polarized light, and reflected normal and UV light microscopy. Different diagenetic alterations (Fig 4) were noted and in some cases semi-quantified, including bioerosion, microcracking, generalized destruction, presence of mineral or organic inclusive material and infiltrations (staining), see Results section for detailed images.

**Table 1. Geoarchaeological methods and their abbreviations.**

| Abbreviation | Method | Description |
|---|---|---|
| MS | Magnetic Susceptibility | Magnetic susceptibility measured on 10g of soil, with a Bartington MS3 system with an MS2B probe [36]. Data are reported as SI-units per ten grams of soil, (corresponding to $X_{lf}$, $10^{-8}$ $m^3$ $kg^{-1}$) [37]. |
| MS550 | Magnetic Susceptibility after burning at 550˚C | Magnetic susceptibility after 550˚ C ignition (units as above) [38]. |
| LOI (%) | Loss On Ignition | Soil organic matter, determined by loss on ignition at 550˚ C, in per cent [39]. |
| Cit-P | Inorganic phosphate content | Extraction with 2% citric acid (mg P/kg dry matter, ppm), corresponding to the Arrhenius method [40]. |
| Cit-POI | Total phosphate | Extraction with 2% citric acid on ignited soil (mg P/kg dry matter, ppm), inorganic & organic [28]. |
| P quota | Cit-POI /Cit-P | The ratio of inorganic & organic to inorganic phosphate |
| pH | | Analysed in 0.1 M KCl solution on wet samples in a 1:5 sample to solution mix |
| XRF | X-ray fluorescence spectroscopy | Thermo Scientific Niton XL5 Analyzer, connected to a Thermo Scientific™ portable test stand. The reference calibration Soil mode was used for quantification |

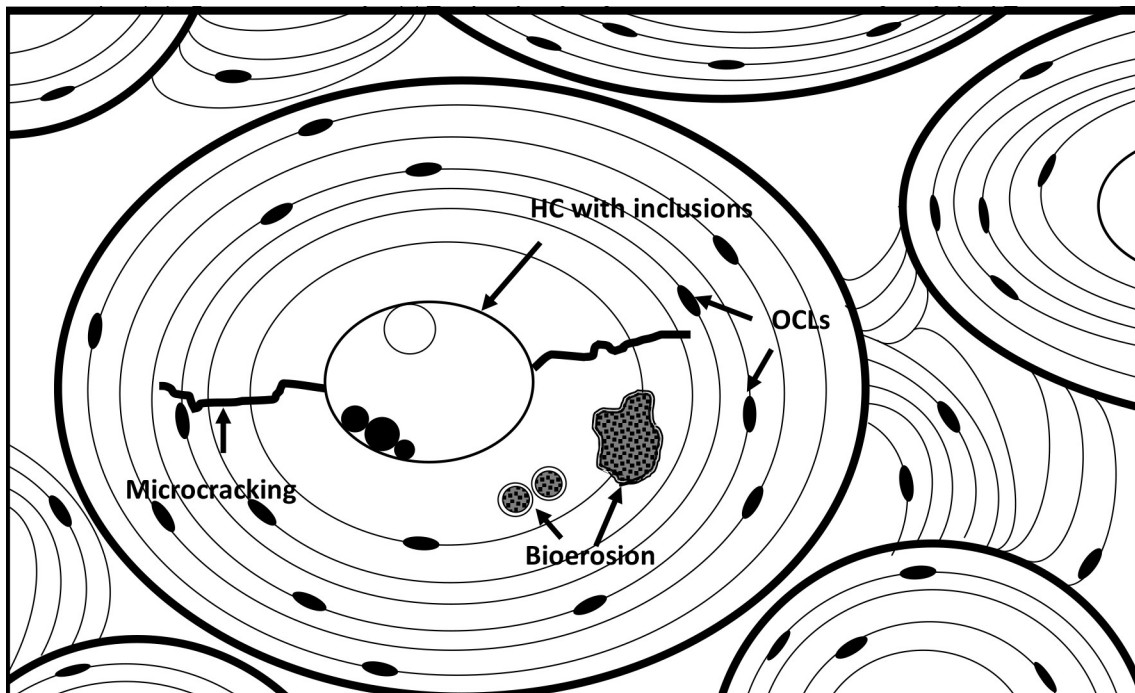

**Fig 4. Schematic drawing showing various microanatomical features of bone.** These include osteons (the bold circles), lamellae (the fine circles/lines), osteocyte lacunae (OCL) and Haversian canals (HC), as well as diagenetic features such as bioerosion, microcracking and inclusive material. Figure by Hege Hollund for this publication.

Generalized destruction is a general loss of microstructural features without any identifiable microbial destruction. Additionally, the extent and intensity of birefringence, that is, the appearance of a pattern of dark and light bands when bone sections are viewed in polarized light, reflects bone preservation, particularly collagen [43, 44]. Bioerosion is identified as distinct alterations to the microstructure in the form of microscopic destructive foci (MFD) first characterized by Hackett [45]. These foci are from five to up to sixty microns across [46] with a mix of demineralized bone, and re-precipitated bone mineral, sometimes with a fine porosity, interpreted as the result of tunnelling by microorganisms, mainly bacteria [47–49]. The extent of this type of bioerosion may be assessed using the so-called Oxford histological index (OHI) [50, 51], scoring the preservation of the bone microstructure on a scale from 5 to 0, where 5 is perfectly preserved and 0 is completely bioeroded. This scoring is made by visual assessment of the approximate percentage intact bone of the sectioned sample, as seen in the light microscope. The OHI is only lowered if bioerosive features can be identified. However, the bone microstructure can be severely damaged by other processes such as generalized destruction, intense cracking and staining. The total amount of damage, by bioerosion and non-biological processes, may be quantified by a parallel preservation index called the General Histological Index (GHI) [52] (see Table 2). This means that a sample with an OHI of 5 suggesting good preservation in terms of bioerosion, may have a lower GHI if other diagenetic processes have led to destruction. If bioerosion is the sole destructive process, the OHI and GHI will be the same. If both bioerosion and other processes have destroyed the microstructure, the GHI will be lower than the OHI. The GHI is only lowered if the alterations have destroyed or are completely obscuring the microstructure as seen in the microscope.

An Olympus BX51 microscope with magnifications of x40–500 was used for the investigations. All analyses were carried out by the same person (Hollund) to avoid inter-observer variation.

**Table 2. Histological indexes used to assess bone preservation.**

| OHI/GHI | % of intact bone | Description |
|---|---|---|
| 0 | 0 | No original features identifiable, except that Haversian canals may be identifiable. |
| 1 | <15 | Small areas of well-preserved bone present |
| 2 | <50 | Some well-preserved bone present between destroyed areas |
| 3 | >50 | Larger areas of well-preserved bone present |
| 4 | >85 | Bone is fairly well preserved with minor amounts of destroyed areas. |
| 5 | 100 | No MFDs/generalized destruction observed |

Descriptions, in terms of the extent of bioerosion (OHI), and all types of destruction, bioerosion and other (GHI), adapted from [51, 52].

## Collagen preservation

Six bone samples from the 2019 excavation were selected for collagen preservation analysis. To this, an additional nine bone samples were taken from the 1940s excavation and three bone samples from the 1970s excavation.

All collagen extractions were made at the Radiocarbon Dating Laboratory, Department of Geology, Lund University, Sweden. The collagen extractions followed a procedure where the bone samples were first mechanically cleaned to remove any superficial stains and non-osseous material. Four of the samples were delivered as a powder and were not subjected to this initial screening by the laboratory; although, prior to sampling, the outer bone surface was removed and discarded before each powder sample was retrieved. The bone samples were then treated with NaOH to remove any remaining humic substances. The ensuing collagen extraction procedure was done following a modified Longin [53] protocol similar to that described in [54]. To ensure uncontaminated archaeological collagen, all samples were subjected to ultrafiltration after HCL treatment. This was done to separate components of high molecular weight (>30 kilodaltons (kD)), representing 'uncontaminated' collagen, from components of low molecular weight (<30 kD), which include degraded collagen, salts and amino acids from the soil etc. [55]. Two of the bone samples from the 2019 excavation campaign were also successfully radiocarbon dated by accelerator mass spectrometry (AMS) using an SSAMS machine [56], also at the Radiocarbon Dating Laboratory, Lund University, following graphitization of the collagen in an AGE-3 automated system [57] coupled to an elemental analyser and calibrated using Oxcal 4.3.

## Palaeobotany

An archaeobotanical analysis was conducted for seven of the 13 soil samples (see soil chemistry), taken from different layers during the excavation, and 0.1 litres of soil per sample was analyzed. The soil was treated with wash-over flotation method to obtain organic remains [58], where the soil samples are put in a beaker where water is added to create an overflow. This enables the organic material to be separated from inorganic material with the overflowing water. The smallest mesh size of 0.4 mm was used for the retrieval of plant material. All material over 1 mm was studied with a stereo microscope with 8–80× magnification, whereas, due to the high organic content making the analysis time-consuming, a cursory analysis was conducted for the 1–0.4 mesh fractions from samples 186 and 191. Seeds and other remains of plants were counted, whereas the presence of insect exoskeletons and earthworm cocoons (*Lumbricus sp.)* were noted. The amount of charcoal and wood was estimated with a relative scale. Plant remains were identified with a reference collection at the Archaeologists in Lund and current literature [59]. ArboDat English version 2018 was used for treatment and storing of data.

A preliminary study of the current local flora at the site was conducted to establish a baseline for possible modern intrusions into the archaeological contexts, which may have occurred due to earthworm movements, dry cracks in the sediment or during the archaeological excavation itself. The species of interest to this study (species which are currently growing in the area with specimens from the species recovered in the soil samples) are hazel (*Corylus avellana*), alder (*Alnus glutinosa*), common nettle (*Urtica dioica*) and raspberry (*Rubus idaeus*).

## Results

### Soil chemistry

One of the most fundamental parameters in understanding archaeological organic preservation is the pH level of the area where the material is buried. At Ageröd, the pH-values range from 4.2 (sample 192) up to 6.7 (samples 191 and 187). This is a large span in the sediment pH, considering the limited size of the sampled area, and indicates highly dynamic sediment chemistry that in the lower pH range result in a rapid deterioration of the deposited bone materials. Interpretations from the soil analyses are based on both wet chemistry (Table 3) and XRF data (Table 4).

In each of the 'dry-soil-sample' trenches, pH generally increases with depth. For the wet areas, we only have one soil sample (sample 188 in trench 209), whereby no stratigraphic analysis can be made (note that while trench 205 is located in 'wetter' zone 3, none of the soil samples were selected from the wet, lower, part of the trench). However, as seen in Fig 2, this does not mean that the trenches are easily comparable to each other as all of the test trenches were dug on the slope down towards the former lake and have different prerequisites depending on their location (e.g. timing of formation, wetness, original peat thickness, subjection to groundwater fluctuation etc.). In zone 2, the thickness of the added soil bank varies depending on where the trench was located, whereby the depth in trench 217 is greater than in 201. Nevertheless, there is a within trench correlation with increasing depths and increasing pH in the dry areas of the excavation. The general increase in pH corresponding with depth is illustrated in Fig 5A–5F, where the vertical variation of pH is shown with the relative amount of different elements.

**Table 3. Result of the wet chemical analyses applied to the different soil samples.**

| Sample nr. | Trench nr. | Zone | Depth (cm) | GW (cm) | Layer | MS | MS550 | MSQ | LOI% | CitP | CitPOI | PQ | pH |
|---|---|---|---|---|---|---|---|---|---|---|---|---|---|
| 180 | 201 | 2 | 40–50 | - | UP | 4 | 760 | 190,0 | 46,2 | 114 | 620 | 5,45 | 5,1 |
| 182 | 201 | 2 | 50–55 | - | UP/CL | 8 | 864 | 108,0 | 19,9 | 204 | 480 | 2,35 | 5,3 |
| 181 | 201 | 2 | 50–60 | - | CL | 7 | 532 | 76,0 | 12,2 | 478 | 474 | 0,99 | 5,9 |
| 183 | 201 | 2 | 60–70 | - | LP | 5 | 799 | 159,8 | 18,1 | 137 | 287 | 2,09 | 6,2 |
| 184 | 205 | 3 | 20–30 | 58 | UP | 5 | 3286 | 657,2 | 50,8 | 103 | 394 | 3,83 | 6,3 |
| 185 | 205 | 3 | 30–40 | 58 | CL | 5 | 1936 | 387,2 | 27,5 | 269 | 478 | 1,78 | 6,5 |
| 186 | 205 | 3 | 40–42 | 58 | CL | 5 | 2199 | 439,8 | 32,5 | 148 | 343 | 2,32 | 6,4 |
| 187 | 205 | 3 | 42–42 | 58 | CL/LP | 30 | 1867 | 62,2 | 17,2 | 51 | 137 | 2,71 | 6,7 |
| 188 | 209 | 3 | 50 | 55 | LP | 14 | 2851 | 203,6 | 39,4 | 54 | 240 | 4,47 | 5,0 |
| 189 | 217 | 2 | 60–70 | - | CL | 5 | 1458 | 291,6 | 44,1 | 91 | 582 | 6,38 | 5,5 |
| 190 | 217 | 2 | 75 | - | CL | 5 | 1385 | 277,0 | 27,7 | 286 | 450 | 1,58 | 6,5 |
| 191 | 217 | 2 | 80–90 | - | LP | 5 | 1760 | 352,0 | 33,5 | 279 | 473 | 1,69 | 6,7 |
| 192 | 259 | 1 | 35 | - | DP | 22 | 364 | 16,5 | 3,9 | 210 | 184 | 0,88 | 4,2 |

GW = groundwater level (depth in cm), UP = Upper peat, CL = white cultural layer, LP = lower peat, DP = degraded peat. Units: MS, MS 550 $X_{lf}$, $10^{-8} m^3 kg^{-1}$; MSQ = MS550/MS; CitP, CitPOI = mg P/kg dry matter, in ppm; PQ = CitPOI/CitP.

**Table 4. XRF-data from the soil analyses of Ageröd I 2019 excavations.**

| Sample | Trench | Zone | Layer | As | Ba | Ca% | Cd | Cr | Cs | Cu | Fe% | K% | Mn | Mo | Ni | Pb | Pd | Rb | S% | Sb | Sn | Sr | Th | Ti | U | V | Zn | Zr |
|---|---|---|---|---|---|---|---|---|---|---|---|---|---|---|---|---|---|---|---|---|---|---|---|---|---|---|---|---|
| 180 | 201 | 2 | UP | 3,4 | 236 | 2,67 | 5,6989 | 58,5 | <LOD | 69,4 | 1,7 | 0,43 | 827 | <LOD | 32,1 | 19,1 | 2,2 | 32,7 | 0,72 | <LOD | <LOD | 123 | 16,0 | 1432 | 39,4 | 80,2 | 143 | 134 |
| 182 | 201 | 2 | UP/CL | 8,3 | 453 | 1,87 | <LOD | 62,4 | 19,2 | 57,2 | 2,9 | 1,05 | 684 | <LOD | 27,9 | 16,9 | <LOD | 52,7 | 0,42 | <LOD | <LOD | 126 | 10,0 | 2503 | 24,3 | 79,5 | 109 | 164 |
| 181 | 201 | 2 | CL | 14,3 | 544 | 4,81 | <LOD | 36,3 | 21,0 | 20,0 | 4,1 | 0,85 | 550 | <LOD | 55,4 | 9,7 | <LOD | 74,4 | 0,85 | <LOD | <LOD | 154 | 5,7 | 1809 | 17,9 | 54,6 | 43 | 183 |
| 183 | 201 | 2 | LP | 15,5 | 344 | 4,24 | <LOD | 42,4 | <LOD | 29,7 | 3,9 | 0,69 | 1916 | <LOD | 68,3 | 12,3 | <LOD | 79,3 | 1,02 | <LOD | <LOD | 124 | 7,2 | 1479 | 27,8 | 79,1 | 49 | 109 |
| 184 | 205 | 3 | UP | 8,8 | 88 | 4,73 | 15,198 | 26,3 | <LOD | 35,9 | 4,6 | 0,3 | 268 | 9,4 | 16,3 | 25,6 | 5,0 | 29,0 | 0,88 | 9,6 | 5,3 | 121 | 7,8 | 578 | 38,2 | 42,7 | 42 | 100 |
| 185 | 205 | 3 | CL | 5,8 | 285 | 3,46 | 4,4614 | 26,1 | <LOD | 11,8 | 4,9 | 0,79 | 288 | 3,4 | 32,5 | 9,1 | 2,0 | 58,9 | 0,75 | <LOD | 5,0 | 134 | 4,8 | 1739 | 23,5 | 45,3 | 11 | 136 |
| 186 | 205 | 3 | CL | 9,2 | 204 | 4,29 | 10,1053 | 19,4 | <LOD | 17,8 | 5,4 | 0,32 | 299 | 6,5 | 37,8 | 5,0 | 3,0 | 38,1 | 0,89 | 7,7 | 5,6 | 123 | 4,0 | 999 | 27,3 | 39,6 | 11 | 113 |
| 187 | 205 | 3 | CL/LP | 11,7 | 397 | 2,81 | <LOD | 27,4 | 18,2 | 15,2 | 5,1 | 0,6 | 446 | 5,6 | 54,2 | 8,5 | <LOD | 63,8 | 0,77 | <LOD | <LOD | 134 | 5,2 | 2024 | 23,0 | 63,9 | 13 | 147 |
| 188 | 209 | 3 | LP | 21,9 | 195 | 2,00 | 7,4616 | 69,6 | <LOD | 59,0 | 6,9 | 0,32 | 198 | 8,0 | 38,4 | 21,9 | 1,3 | 34,7 | 1,74 | 5,8 | 3,2 | 82 | 9,1 | 1258 | 34,0 | 99,0 | 15 | 120 |
| 189 | 217 | 2 | CL | 7,9 | 215 | 2,96 | 4,0592 | 32,2 | <LOD | 71,5 | 2,6 | 0,42 | 1430 | <LOD | 42,5 | 45,4 | <LOD | 32,9 | 0,77 | <LOD | <LOD | 106 | 10,8 | 1768 | 32,6 | 72,1 | 169 | 146 |
| 190 | 217 | 2 | CL | 10,7 | 286 | 6,14 | <LOD | 12,9 | <LOD | 37,5 | 3,8 | 0,45 | 950 | <LOD | 58,5 | 4,9 | <LOD | 40,1 | 1,29 | <LOD | <LOD | 148 | 5,8 | 1279 | 24,5 | 66,5 | 60 | 124 |
| 191 | 217 | 2 | LP | 16,0 | 208 | 5,85 | 2,8197 | 10,7 | <LOD | 23,0 | 4,3 | 0,46 | 1203 | <LOD | 55,6 | 3,9 | <LOD | 34,7 | 1,09 | <LOD | <LOD | 143 | 4,2 | 1283 | 19,9 | 54,1 | 54 | 100 |
| 192 | 259 | 1 | DP | 7,7 | 621 | 0,66 | <LOD | 34,2 | 44,3 | 5,2 | 2,9 | 1,69 | 320 | <LOD | 32,4 | 22,8 | <LOD | 88,7 | 0,07 | <LOD | <LOD | 138 | 7,7 | 3188 | 3,0 | 51,5 | 59 | 209 |

Data reported as mg*kg⁻¹ unless given in %. Up = Upper peat, CL = white cultural layer, LP = Lower peat, DP = Degraded peat.

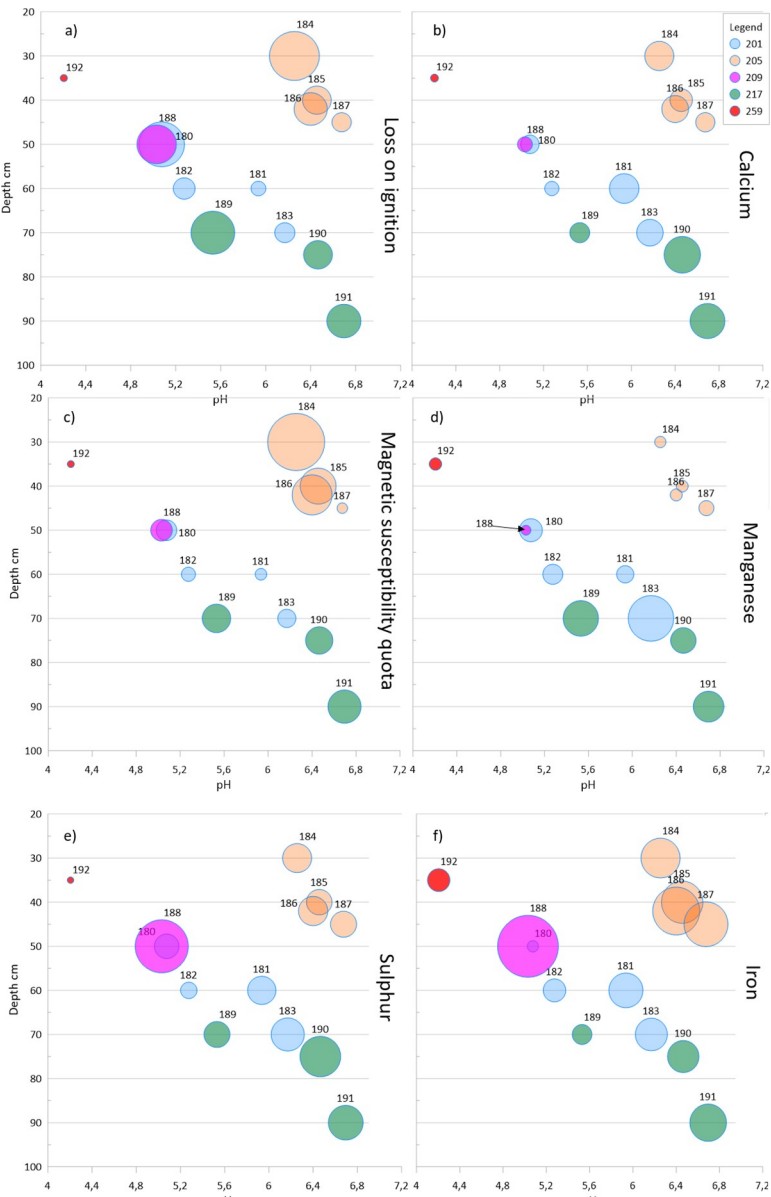

**Fig 5. Variations in soil chemistry as a function of pH and depth.** The size of the dots reflect the relative amount of each component in the analysed sample and the colour of the dot refers to a trench. a) Organic matter (LOI, range: 3.9–50.8%), b) Calcium (Ca, range: 0.7–6.1%), c) Magnetic susceptibility quota (MSQ, range: 16.5–657.2), d) Manganese (Mn, range: 198–1916 ppm), e) Sulphur (S, range: 0.07–1.74%), f) Iron (Fe, range: 1.7–6.9%).

Fig 5 shows pH as a function of the depth within each trench with the magnitude of the varied soil chemistry parameters separately plotted. Fig 5A shows the variation of organic matter (LOI) in the different soil samples and Fig 5B the relative amounts of calcium (Ca). The correlation between pH and trench depth is further related to the MS-quota (Fig 5C), which can act as a proxy for the shifting redox potential via the $Fe^{II}$ to $Fe^{III}$ system (iron oxide, FeO, to iron oxide-hydroxide, FeO(OH), from reducing to oxidizing conditions [28, 60]. The quota of MS550 to MS indicates the state of iron, and the higher the MSQ the more initial $Fe^{II}$ is present in the system and thus points to reducing conditions. The most degraded sample, 192, has the lowest MSQ whereas the highest is represented by sample 184. However, this is also related to

the actual amount of organic matter and the fact that the conversion of $Fe^{II}$ to $Fe^{III}$ is more efficient the more organic matter is present in the sample. Trench 205 stands out and is represented with high MSQ, apart its lowermost soil sample which has a low MS-quota. There are clear differences between the trenches 201 and 217 even though they belong to the same zone of sampling and are both found under the soil bank from the drainage ditch dug in the early 20th century. The observed differences are likely caused by the latter being sampled at 70 cm depth and lower while the first is sampled between depths of 50–70 cm (the same archaeological layers are sampled but the soil bank cover was thicker in trench 217, see Fig 3).

Fig 5D shows the manganese (Mn) content in a similar way. Normally, Mn would migrate towards higher pH in a soil-sediment system. In zone 3, trenches 205 and 209, all samples (184–188) are depleted from Mn, something which is likely caused by fluctuations in the water table (see below). The same level of Mn depletion is observable in trench 259, although since this trench is located further up the slope, this depletion is likely caused by increased degrading conditions from the soil having completely oxidized, possibly also with an increased depletion caused by the previous undocumented disturbance in the trench.

The relative amounts of sulphur (S) (Fig 5E) is also an important factor for determining the soil chemical properties. Trench 259, sample 192, stands out with low relative amounts of S in a completely oxidised soil. On the other end of the scale trench 209, sample 188, appears with the highest reading of S. Looking at the iron (Fe) content here (Fig 5F), the presence of oxidized pyrite may be an explanation to this pattern (see Bone histology). Sample 188 has the highest amounts of S but is found in the lower pH range; which may represent an initial and rapid phase of decay of the organic matter, as the sulphur is not yet leached. Because trench 209 displays both high S and Fe values it might be related to the pyrite and a currently ongoing process, leading to a drop in pH when the soil becomes oxidised. Sample 192 in trench 259 may already have reached its lowest pH, looking at the S-Fe relation. Whereas in trench 209, it seems to be an active degradation process that has gone on for some time, which is showing in the low pH level of the trench. However, here a further drop in pH may follow caused by a still ongoing oxidation process.

To obtain more information hidden in this multivariate data set and thus in the sediment system a principal component analysis (PCA) was conducted [61]. Here, the main directions for the sample objects (score t values) are explored along with the corresponding variable loadings (p-values), which makes it possible to observe how samples and variables relate to each other both externally and internally. The produced PC-model gives five significant principal components explaining 94.5% of the total variation of this data matrix. The model and the subsequent figures (Figs 6 and 7) shows the objects similarity in composition, the correlation between the variables and it suggest trends and directions of changes in the sediment system.

Fig 6 shows the sediment characteristics of the samples in terms of MS, phosphate, organic matter, pH and XRF data. The phosphate content is used for assessing the human impact on the archaeological deposit, as the phosphate content to a certain degree follow the impact and intensity during the formation of the layers. Also, phosphate fractionation is used for investigating the relation of inorganic to organic phosphate in a decomposing peat system. The XRF data in the model in Fig 6 shows how the lithogenic elements (rubidium (Rb), zirconium (Zr), potassium (K) and titanium (Ti)) relate to the negative p1 values. Chalcophile elements (copper (Cu), lead (Pb) and zinc (Zn)) form a cluster in the positive p1-p2 axis. Arsenic (As), however, does not seem to follow the expected trend. Instead Fe, Nickel (Ni), As and S follow the negative p2 loading axis. The presence of pyrite is likely causing this pattern.

In zone 3 (profile 205 and 209), the Mn content is generally lower, and Fe consequently shows higher concentrations here. This should be an example of shifting water tables and lateral water movement under reducing conditions leading to loss of Mn [62]. Also, the plot indicates which samples that are most likely to have better-buffering capacity than the others.

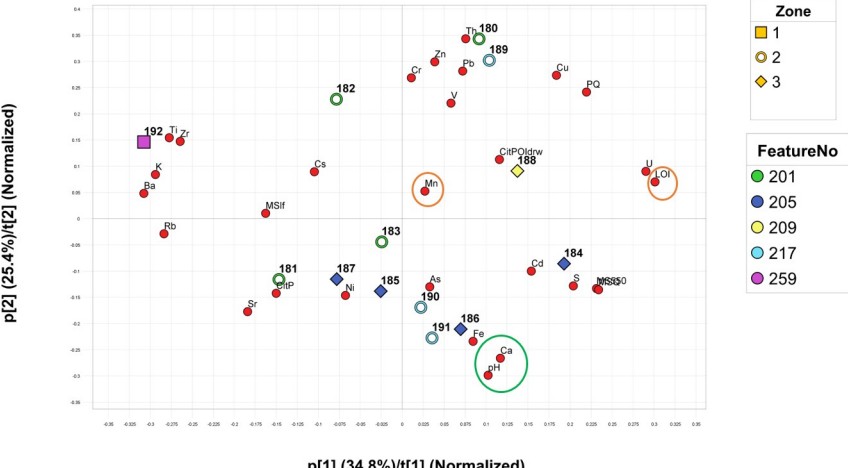

**Fig 6. PC plot (t1 vs t2).** Model based on the analysed parameters (phosphates, Magnetic susceptibility, Loss on ignition and pH) and XRF-data set. Green encircled variables indicate buffer (pH and Ca) and orange encircled variables indicate reducing conditions (LOI and Mn).

Fig 7 gives the p2t2 to p3t3 score-loading plot. Here the most important observation is the negative p3-t3 orientation of samples 187, 188 and 192, as they are turning up as trending to a potentially rapid shift of pH and decomposition of organic materials (followed by samples 184 and 186, also from trench 205, which are also on the negative p3-t3 axis).

## Bone histology

Overall there is little bioerosion present in the studied sample assemblage considering microscopic focal destruction (MFD), and only four out of the fourteen samples display a lowered OHI value (see Table 5) of 3–4 while one sample is completely bioeroded with an OHI of 0. All samples, however, displayed enlarged osteocyte lacunae (bone cell pores) and canaliculi, the interconnecting channels between the lacunae, which may be attributed either to etching,

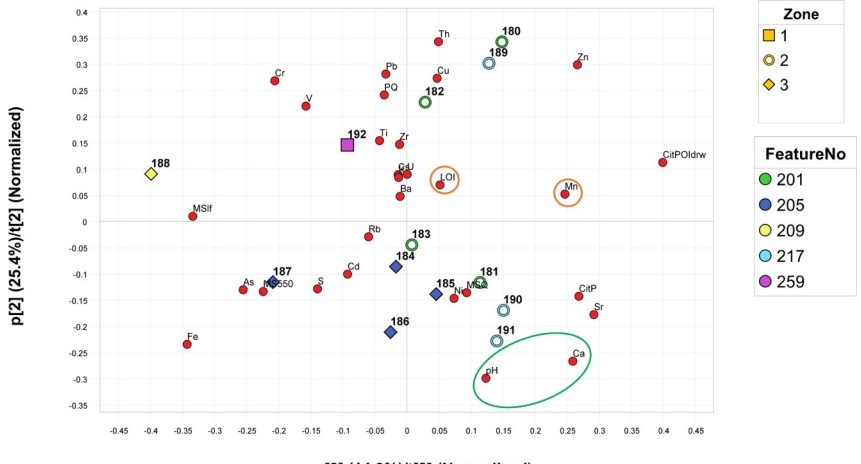

**Fig 7. PC plot (t3 vs t2).** Model based on the analysed parameters (phosphates, Magnetic susceptibility, Loss on ignition and pH) and XRF-data set. Green encircled variables indicate buffer (pH and Ca) and orange encircled variables indicate reducing conditions (LOI and Mn).

**Table 5. General information concerning the histologically analysed bone samples.**

| IDnr | Year | Trench | Zone | Layer | Coordinate | Species | Anatomy | OHI | GHI | Inclusions | Weathering |
|------|------|--------|------|-------|------------|---------|---------|-----|-----|------------|------------|
| A1 | 1940 | | 3 | CL | x-1/y+51 | Wild boar (*Sus scrofa*) | Calcaneus | 5 | 3 | Pyrite | 2 |
| A2 | 1940 | | 3 | CL | x-2/y+54 | Red deer (*Cervus elaphus*) | Radius | 5 | 2 | | 2 |
| A3 | 1940 | | 3 | CL | x-2/y+51 | Roe deer (*Capreolus capreolus*) | Femur | 3 | 4 | Pyrite | 2 |
| A4 | 1970 | | 2 | LP | nd | Elk/Moose (*Alces alces*) | Femur | 5 | 4 | Pyrite | 3 |
| A5 | 1970 | | 2 | CL | nd | Red deer (*Cervus elaphus*) | Scapula | 5 | 3 | | 3 |
| A6 | 1940 | | 2 | CL | x-8/y+26 | Aurochs (*Bos primigenius*) | Metatarsus | 0 | 0 | | 4 |
| ID6 | 2019 | 217 | 2 | CL | x-7/y+21 | Aurochs (*Bos primigenius*) | Metatarsus | 3 | 2 | Ox. pyrite | 4 (6) |
| ID9 | 2019 | 217 | 2 | LP | x-7/y+21 | Elk/Moose (*Alces alces*) | Metacarpus | 4 | 3 | Ox. pyrite | 3 |
| ID56 | 2019 | 205 | 3 | LP | x-1/+45 | Wild boar (*Sus scrofa*) | Calcaneus | 5 | 2 | Ox. pyrite | 4 (8) |
| ID62 | 2019 | 205 | 3 | CL | x-1/+45 | Red deer (*Cervus elaphus*) | Tibia | 5 | 2 | Ox. pyrite | 4 (7) |
| ID70 | 2019 | 201 | 2 | UP | x-6/y+47 | Brown bear (*Ursus arctos*) | Coxae | 5 | 1 | | 4 (6) |
| ID81 | 2019 | 201 | 2 | CL | x-6/y+47 | Elk/Moose (*Alces alces*) | Radius | 5 | 1 | | 3 |
| ID107 | 2019 | 201 | 2 | LP | x-6/y+47 | Wild boar (*Sus scrofa*) | Radius | 5 | 1 | | 4 (6) |
| ID114 | 2019 | 201 | 2 | LP | x-6/y+47 | Roe deer (*Capreolus capreolus*) | Femur | 5 | 3 | Ox. pyrite | 3 |

Weathering analyses from [1] (severity of degradation increases with increasing weathering numbers).

nd = no data. Up = Upper peat, CL = white cultural layer, LP = Lower peat.

staining and/or bioerosion. In some cases broadening and lengthening, and branching of canaliculi is clear, similar to MFDs that have been termed Wedl type 2 [63]. The OHI has not been lowered when such features were observed as it is not clear whether or not bioerosion is involved.

All samples have lowered general histological index due to generalized destruction, extensive cracking and dark staining, some severely with a GHI of 1 despite no identifiable bioerosion. This affects the birefringence which in some samples is reduced to small areas across the middle cortex (Fig 8). The most intense staining is generally found along surfaces, but whole sections display yellowish and brownish staining, in larger areas, or smaller spots of brown, orange or reddish colour (Fig 9). These likely include mineral compounds containing iron and manganese, and organic humic substances infiltrating the bone.

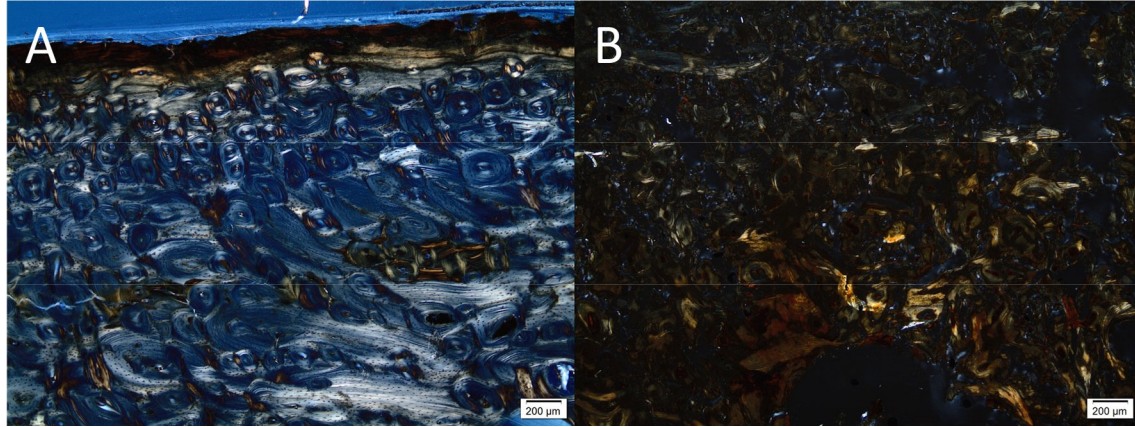

**Fig 8.** Micrograph of sample A1 (A) and ID70 (B), excavated in 1940 and 2019 respectively, in polarized light. Sample A1 display bright birefringence across the whole sample except a narrow band along the outer surface, whereas sample ID7 hardly displays any birefringence at all and the image appears dark. Photo by Hege Hollund for this publication.

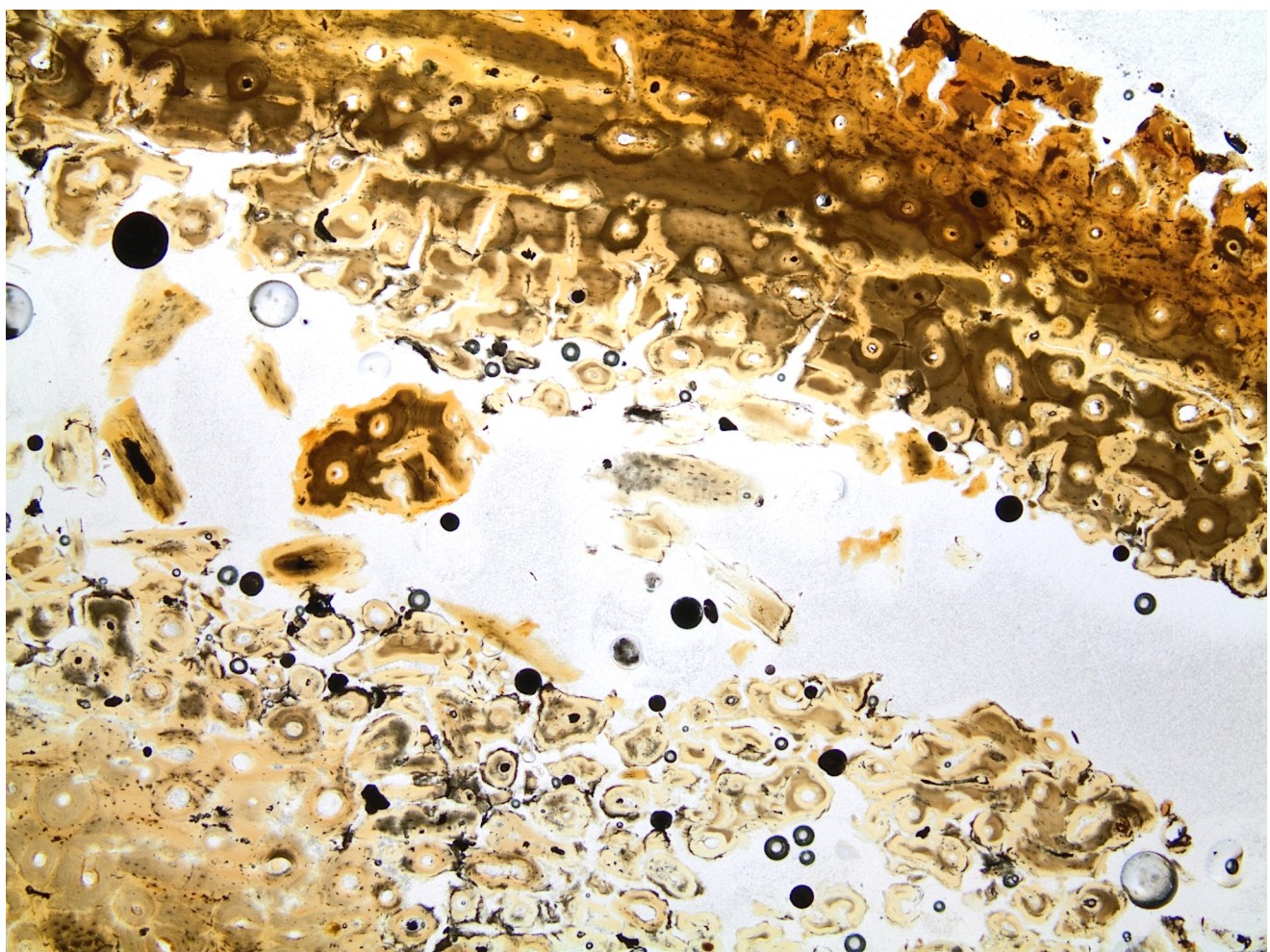

**Fig 9. Micrograph of sample ID62, excavated in 2019, displaying yellow, brown and orange staining across the whole thickness of the bone.** There are large cracks across the middle cortex, likely exacerbated by the sample preparation. Photo by Hege Hollund for this publication.

Inclusive material is present in all samples. The osteocyte lacunae, vascular canals and cracks are filled with opaque matter, or material similarly coloured to that of the staining. In eight of the bones, some of these inclusions could be identified as framboidal pyrite, which are spherical clusters of iron sulphide ($FeS_2$) crystals (Fig 10). These generally appear opaque in transmitted light, and bright metallic in reflected light. Pyrite only forms under anoxic conditions but may oxidize to form iron oxyhydroxide and sulphuric acid if oxygen is introduced into the environment [52, 64]. This has happened in sample ID114, as the grains appear orange in a transmitted light microscope (Fig 10C and 10D). In the other samples from the 2019 excavation, the pyrite was not evident when studying the samples in transmitted light as the pores seemed filled with dark, fuzzy material and no framboidal pyrite shapes could be seen. Only upon investigation in reflected light did it become apparent that some of this material was partially oxidized pyrite grains, often with an intact core and an oxidized outer rim. Conversely, three of the six samples from the older excavations contained intact framboidal pyrites suggesting stable, anoxic conditions throughout the burial period (Fig 10A and 10B).

All samples, both from previous and the most recent excavation campaign, also contain inclusions in the form of transparent, grey shapes, mostly spherical, within large pores or

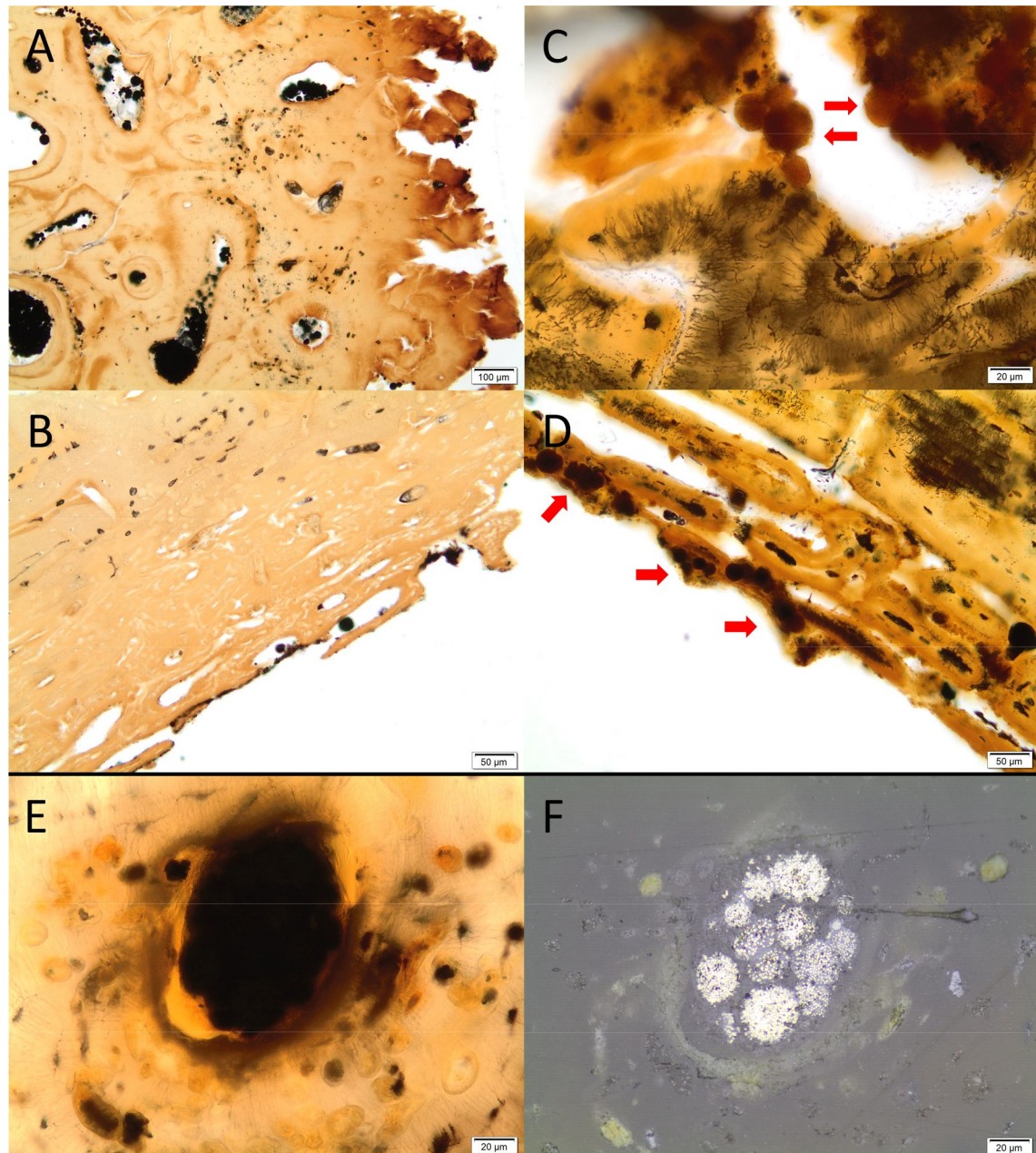

**Fig 10. Micrographs of bone samples displaying grains of pyrite and oxidized pyrite within bone samples.** (A-B) Two samples (A1 and A3) from the 1940s contain numerous intact pyrite grains within bone pores, appearing as black/opaque spheres (Samples A1 and A3). C-D) A sample excavated in 2019 (ID114) contain oxidized pyrite grains, retaining the shape but appearing reddish-brown and translucent in normal transmitted light. E-F) Sample ID9 excavated in 2019, in transmitted (E) and reflected light (F), the latter showing that the blurry mass seen in transmitted light contains pyrite grains. Photo by Hege Hollund for this publication.

cracks in the bone, and on the surface or in the resin immediately surrounding the samples. Some of these are likely fungal structures, and some may be budding fungal cells, as well as possible biofilm, supported by the fact that these features displayed fluorescence when viewed in UV light (Fig 11A–11F). The placement on/by the surface, and the fact that these organic structures do not appear mineralized by iron or manganese compounds, suggest that they represent relatively recent microbial growth. It is difficult to say what this reflects, the burial

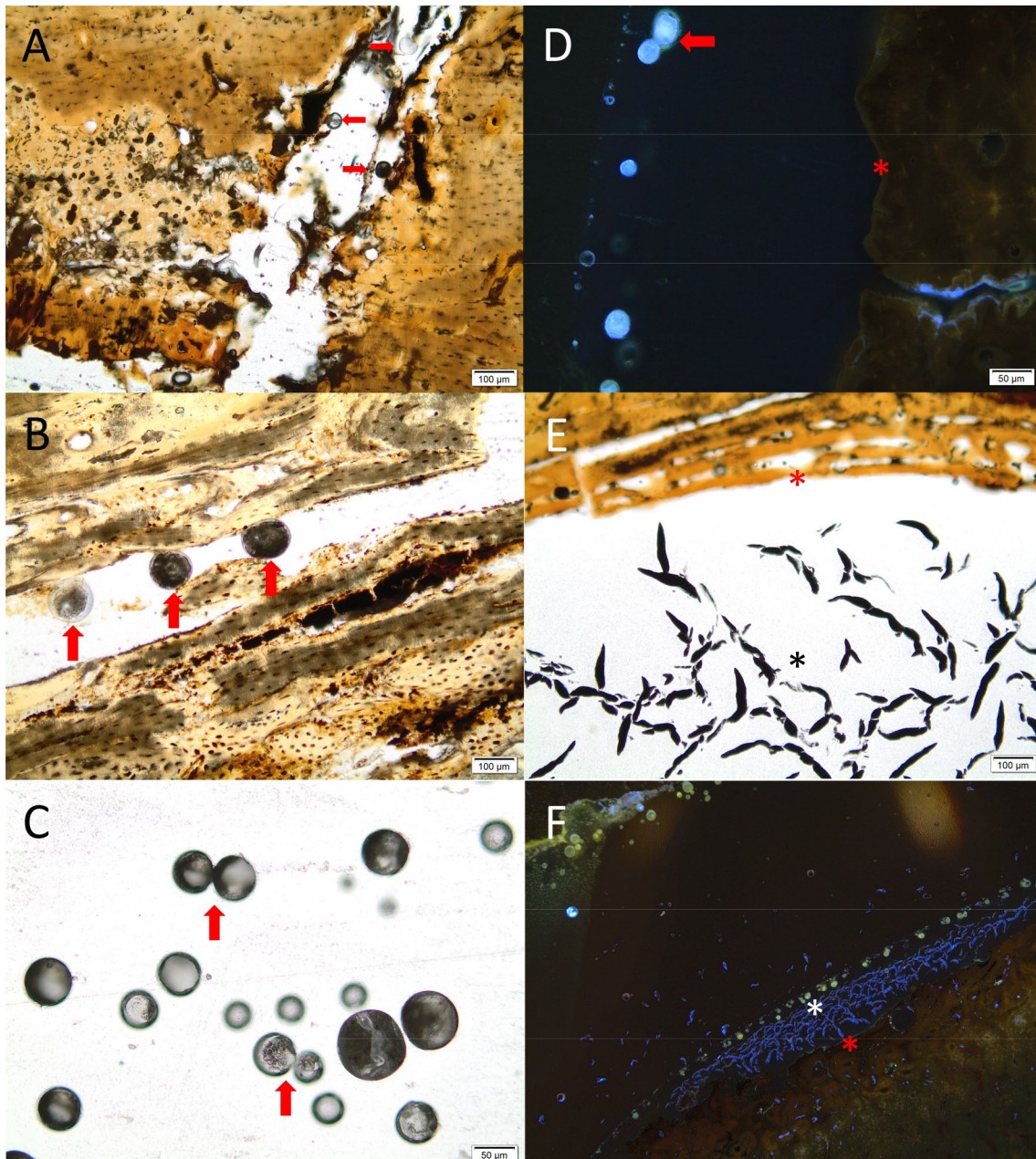

**Fig 11. Micrographs of bone samples with possible recent microbial growth with apparent fungal structures and biofilm within pores, on the surface and 'floating' in the resin the sample is embedded in.** A-B) Sample ID9 and ID81 with spherical, grey and transparent spheres within cracks and pores (red arrows). These could be fungal fruiting bodies. C) The same spherical shapes floating in the resin above the surface of sample ID81, where two shapes (red arrows) look like budding (dividing) fungal cells. D) Sample ID9 seen in UV-light, with a similar shape of a budding cell (red arrow) in the resin directly above the bone surface (red asterisk), with a light blue fluorescens. E) dark, elongated shapes near the endosteal surface of ID114, possibly biofilm pushed off the surface during sample preparation. F) The same elongated shapes (asterisk) observed in UV-light on the periosteal surface (red asterisk) of sample ID114. Photo by Hege Hollund for this publication.

environment, or the post-excavation storage environment. Limited research has been carried out on post-excavation microbial growth, but some studies suggest that most of the microbiota on archaeological bone stem from the burial environment [65–67]. In this case, oxidation of

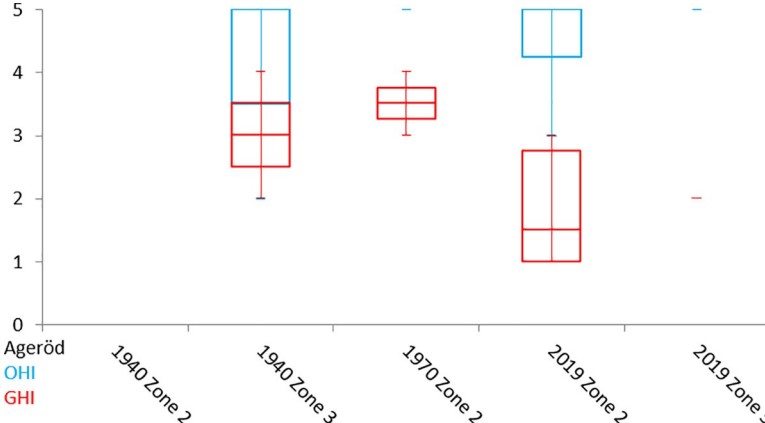

**Fig 12. OHI and GHI values for the histologically analysed bones from Ageröd.** Divided into excavation campaigns and zones.

the burial environment may have caused a proliferation in microbes, while the post-excavation conditions accelerated growth of microbial species already present upon excavation.

The 2019 material has lower average GHI (Fig 12), and visually appears more severely stained, cracked and etched, with low birefringence. The bioeroded samples are primarily from the white cultural layer, whereas pyrite is primarily found in samples from the lower peat layers. The best-preserved sample according to the histological observations is an elk (moose in North America) femur (A4) from the lower peat layer in zone 2, excavated in the 1970s. The worst preserved sample is an aurochs metatarsal (A6) from the white cultural layer in zone 2, excavated in the 1940s (Table 5).

The low number of samples does not allow any robust statistical analyses and if a larger number of bones from the 1940s and 1970s excavations had been analysed, a better histological profile would have been obtained. Such bones as A6 would then not have come to define the histology of a whole zone, and the observations gained from singular bones would be limited, i.e. as illustrated by the weathering analyses where 3444 bones from the three excavation campaigns were used to quantify the status of the osseous remains [1].

Nevertheless, some observations are worth noticing. The preservation and variation in diagenetic traits seem to be related to the location at the site as well as the time of excavation. The bone samples from zone 3 seem to have very little bioerosion, indicating limited microbial activity mainly along the outer surfaces. However, these same bones from trench 205 in zone 3 have been extensively affected by general destruction such as surface corrosion, etching and cracking. This suggests that external, non-biological forces relating to reducing and/or fluctuating redox condition, have affected the bones in zone 3.

## Collagen preservation

The collagen preservation is diverse and the amount of extracted collagen varies depending on species, bone element, sampling method, archaeological deposition layer and excavation campaign (Table 6).

Based on the zooarchaeological analysis of the bone remains from the site [1], the collagen preservation was suspected to correspond with the results from the zooarchaeological analysis, whereby all bone remains intended for collagen preservation studies were sampled from the area of best bone preservation, zone 2, under the soil bank. The potential to also acquire a chronological stratigraphic sequence from the site prompted the use of trench 201, as it was

**Table 6. Results of the collagen preservation analyses.**

| ID | Excavation | Species | Bone | Zone | Coordinate/ trench | Layer | Sample | Sample size (mg) | Sample size after mechanical cleanup (mg) | Sample size after NaOH (mg) | Amount of collagen after HCl and ultra filtration (mg) | % collagen (coll/sample after mechanical) | Weathering degree |
|---|---|---|---|---|---|---|---|---|---|---|---|---|---|
| A I:1 | 1940s | Aurochs | Calcaneus | 3 | X:-1 Y:+48 | LP | Powder | 474,8 | N.A. | 246,8 | 0,7 | 0,15% | 4 |
| A I:2 | 1940s | Aurochs | Calcaneus | 3 | X:-1 Y:+53 | CL | Powder | 560,5 | N.A. | 260,0 | 0,1 | 0,02% | 2 |
| A I:5 | 1940s | Aurochs | Metatarsal | 2 | X:-6 Y: +51 | LP | Powder | 505,4 | N.A. | 207,6 | 0,1 | 0,02% | 3 |
| A I:8 | 1940s | Aurochs | Humerus | 1 | X:-9 Y:+52 | CL | Powder | 576,4 | N.A. | 312,8 | 1,6 | 0,28% | 2 |
| A I:9 | 1940s | Aurochs | Humerus | 3 | X:-1 Y:+27 | CL | Bone piece | 909,8 | 750,0 | 578,1 | 22,7 | 3,03% | 3 |
| A I:17 | 1940s | Elk/ Moose | Radius | 1 | X = -9 Y:+25 | CL | Bone piece | 1340,0 | 1250,0 | 1020 | 1,7 | 0,14% | 3 |
| A I:18 | 1940s | Elk/ Moose | Calcaneus | 3 | X = -0 Y = +67 | CL | Bone piece | 863,6 | 781,0 | 626,6 | 12,7 | 1,63% | 3 |
| A I:23 | 1940s | Red deer | Metacarpal | 3 | X-1 Y: +48 | LP | Bone piece | 510,3 | 489,0 | 360,0 | 11,7 | 2,39% | 2 |
| A I:26 | 1940s | Red deer | Metacarpal | 2 | X:-6 Y:+51 | CL | Bone piece | 1400,0 | 1320,0 | 1060 | 25,2 | 1,91% | 2 |
| A I:13 | 1970s | Elk/ Moose | Humerus | 2 | No data | UP | Bone piece | 1040,0 | 923,0 | 665,2 | 5,2 | 0,56% | 4 |
| A I:21 | 1970s | Red deer | Humerus | 2 | No data | LP | Bone piece | 693,0 | 657,0 | 456,1 | 6,6 | 1,00% | 4 |
| A I:22 | 1970s | Red deer | Radius | 2 | No data | CL | Bone piece | 581,0 | 543,0 | 365,0 | 3,1 | 0,57% | 4 |
| ID 69 | 2019 | Elk/ Moose | Phalanx 2 | 2 | 201 | UP | Bone piece | 1340,0 | 985,0 | 758,4 | 0,9 | 0,09% | 6 |
| ID 70* | 2019 | Brown bear | Coxae | 2 | 201 | UP | Bone piece | 502,7 | 502,0 | 213,3 | 0,0 | 0,00% | 6 |
| ID 81* | 2019 | Elk/ Moose | Radius | 2 | 201 | CL | Bone piece | 877,9 | 875,0 | 603,0 | 0,0 | 0,00% | 3 |
| ID 82 | 2019 | Ungulate | indet. diaphys | 2 | 201 | CL | Bone piece | 1890,0 | 1430,0 | 1170 | 0,0 | 0,00% | 3 |
| ID 107* | 2019 | Wild boar | Radius | 2 | 201 | LP | Bone piece | 637,6 | 636,0 | 465,1 | 2,3 | 0,36% | 6 |
| ID 114* | 2019 | Roe deer | Femur | 2 | 201 | LP | Bone piece | 401,7 | 399,0 | 293,0 | 4,0 | 1,00% | 3 |

*shows that the sample was also histologically analysed, see Table 5.

deemed best suited for these combined purposes; whereby, all bone samples used to test collagen preservation were selected from this trench. However, only three of the six bone samples from the 2019 excavation campaign sent to the radiocarbon lab had any preserved collagen. From those three fragments it was possible to graphitize and radiocarbon date two samples, which suggests that the extracted collagen from the third fragment was, even though ultrafiltered, of poor quality or contaminated. Both datable samples came from the lower peat and gave Middle Mesolithic results (8589–8192 cal BP), corresponding with data from previous radiocarbon dating at the site (Fig 13).

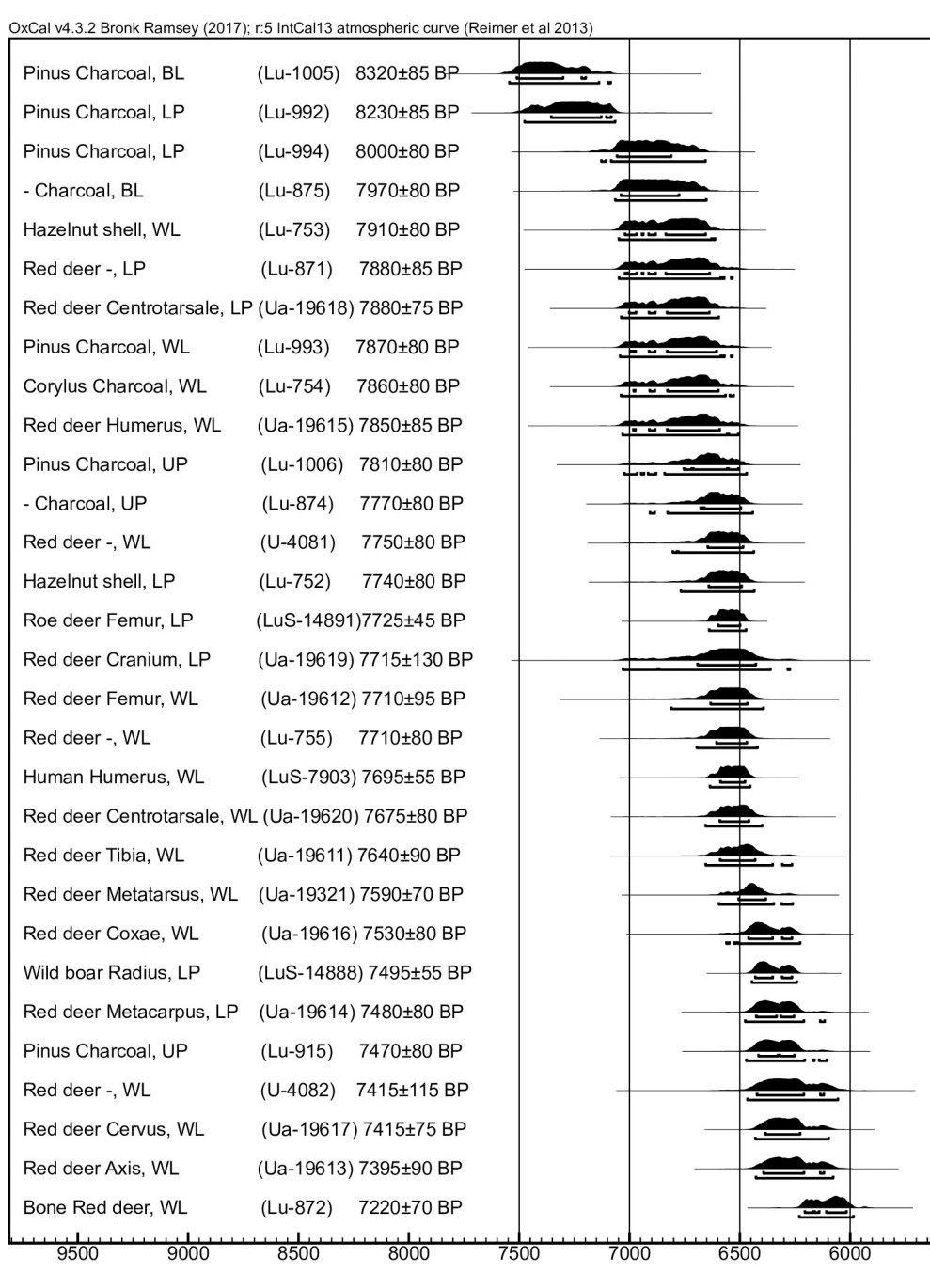

OxCal v4.3.2 Bronk Ramsey (2017); r:5 IntCal13 atmospheric curve (Reimer et al 2013)

**Fig 13. Collation of all radiocarbon dates done on charcoal, hazelnut shells and bones from Ageröd I:HC.** Data from Larsson (1978), Magnell (2006) and previously unpublished data LuS-7903. BL = Bottom layer, LP = Lower peat, UP = Upper peat, CL = white cultural layer. LuS-14888 and LuS-14891 are the wild boar radius respectively roe deer femur from the 2019 excavation campaign.

To understand ongoing changes in collagen preservation, boxplots of the collagen yield from each excavation campaign were created. The largest amount of collagen could be obtained from the bones from the 1940s, although the span is large due to the four samples handed in as powder, as they yielded significantly less collagen compared to the 'bone piece'

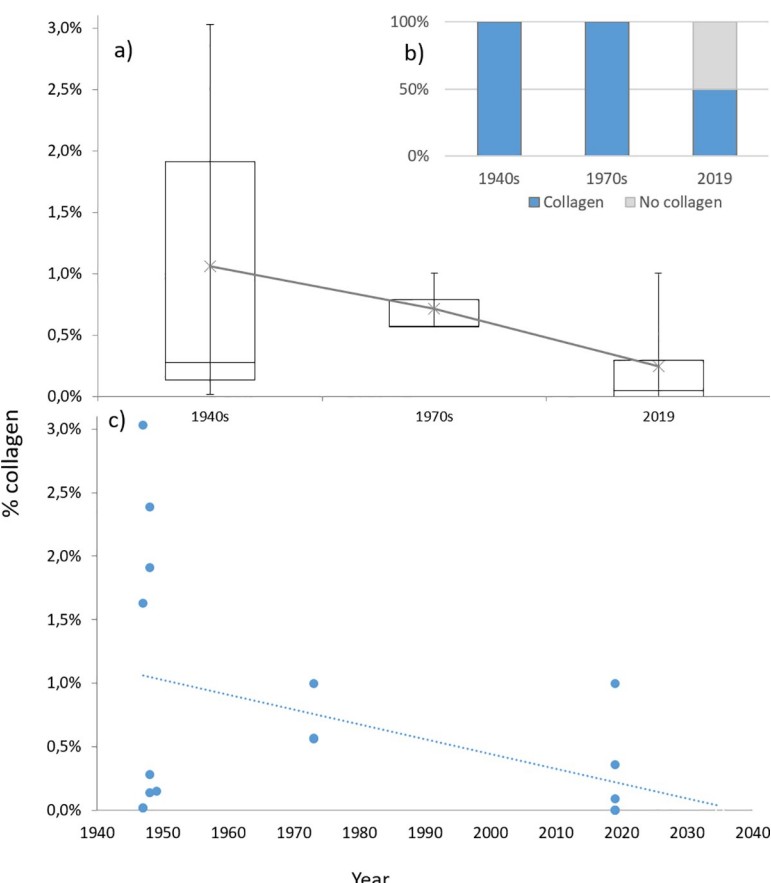

**Fig 14. Comparison of collagen preservation between the different excavation campaigns.** a) Boxplot of the amount of preserved collagen, illustrated as the median of the collagen yield percentage from each excavation campaign with the upper and lower quartiles added, with whiskers added to include the outliers. Average value added as an X with a trend line connecting the different excavation campaigns. b) Illustration of collagen preservation showing that all bone samples from the 1940s and the 1970s had preserved collagen while only half of the bone samples from 2019 had preserved collagen. c) the data inserted into a chronological frame showing a hypothetical trend line when, if no preservative actions are taken and the organic deterioration continues at the same rate and is not further accelerated, collagen might no longer be preserved at Ageröd.

samples. From the 1940s there is a downwards trend, with diminishing collagen yield from the samples (Fig 14A). This trend is further highlighted as all of the samples from the 1940s and 1970s excavation campaigns yielded collagen, while only 50% of the samples from 2019 had any collagen preserved (Fig 14B), which suggest that we might soon be unable to conduct molecular level analyses on bone remains from Ageröd (Fig 14C).

## Palaeobotany

The seven archaeobotanically analysed soil samples contained 87 identifiable waterlogged remains of plants (Table 7). The plant material was generally well preserved and even soft plant tissue, such as catkin bracts, were found preserved in the samples from zone 3.

As seen in Table 7, most of the palaeobotanical material derives from the zone 3 trenches. Trench 209 is the only sample collected from wet conditions and thus holds almost half of the recovered palaeobotanical remains.

The palaeobotanical remains (Fig 15) suggest a shoreline deposition environment within a largely broad-leaved forest. The relatively large amount of edible plant remains such as,

**Table 7. Archaeobotanical analysis of seven soil samples from Ageröd.**

| | | | Sample number | 182 | 183 | 189 | 191 | 186 | 187 | 188 |
|---|---|---|---|---|---|---|---|---|---|---|
| | | | Trench | 201 | 201 | 217 | 217 | 205 | 205 | 209 |
| | | | Zone | 2 | 2 | 2 | 2 | 3 | 3 | 3 |
| | | | Layer | UP/CL | LP | CL | LP | CL | CL/LP | LP |
| | | | Studied volume (l) | 0,1 | 0,1 | 0,1 | 0,1 | 0,1 | 0,1 | 0,1 |
| **English name** | **Latin name** | **Type** | Nr. of remains | 3 | 1 | 11 | 3 | 22 | 7 | 40 |
| | | | Nr. of taxa | 2 | 1 | 7 | 3 | 5 | 5 | 10 |
| **Shoreline vegetation** | | | **Preservation** | | | | | | | |
| Grass-leaved Orache | *Atriplex littoralis* | Seed/Fruit | Waterlogged | | | 1 | | | | |
| White Water-lily | *Nymphaea alba* | Seed/Fruit | Waterlogged | | | | | | 1 | |
| Water-Pepper | *Persicaria hydropiper* | Seed/Fruit | Waterlogged | | | | | | | 7 |
| Pale Persicaria | *Persicaria lapathifolia* | Seed/Fruit | Waterlogged | | | | 1 | | | 4 |
| Pond Weed | *Potamogeton* sp. | Seed/Fruit | Waterlogged | | | | | 3 | 1 | |
| Common Club-rush | *Scirpus* cf. *lacustris* | Seed/Fruit | Waterlogged | | 1 | | | | | 1 |
| **Meadow vegetation** | | | | | | | | | | |
| Marsh Woundwort | *Stachys palustris* | Seed/Fruit | Waterlogged | | | | | | | 1 |
| **Ruderal vegetation** | | | | | | | | | | |
| Shepherd's-purse | cf. *Capsella bursa-pastoris* | Seed/Fruit | Waterlogged | | | | | 1 | | |
| Hemp Nettle | *Galeopsis* sp. | Seed/Fruit | Waterlogged | | | | | | | 1 |
| Common Nettle | *Urtica dioica* | Seed/Fruit | Waterlogged | 2 | | 3 | | | | 1 |
| **Broad leaved forest** | | | | | | | | | | |
| Alder | *Alnus* sp. | Seed/Fruit | Waterlogged | 1 | | 2 | 1 | | | |
| Alder | *Alnus* sp. | Cone Scale | Waterlogged | | | | | | 1 | |
| Silver/Downy Birch | *Betula pendula/pubescens* | Seed/Fruit | Waterlogged | | | | | 1 | | |
| Silver/Downy Birch | *Betula pendula/pubescens* | Catkin Bract | Waterlogged | | | | | | 2 | |
| Hazel | *Corylus avellana* | Seed/Fruit | Waterlogged | | | | | 15 | | 22 |
| Crab Apple | *Malus sylvestris* | Seed/Fruit | Waterlogged | | | 1 | | 2 | | |
| Aspen | *Populus tremula* | Catkin Bract | Waterlogged | | | | | | 2 | |
| Raspberry | *Rubus idaeus* | Seed/Fruit | Waterlogged | | | 1 | | | | 1 |
| **Varia** | | | | | | | | | | |
| Crucifers | Brassicaceae | Seed/Fruit | Waterlogged | | | | | 1 | | |
| Sedge | *Carex* sp. | Seed/Fruit | Waterlogged | | | 2 | | | | |
| Grass | Poaceae | Seed/Fruit | Waterlogged | | | 1 | | | | 1 |
| Creeping Buttercup | *Ranunculus repens* | Seed/Fruit | Waterlogged | | | | | | | 1 |
| Indeterminata | Indeterminata | Seed/Fruit | Waterlogged | 1 | | 1 | | | | |

(*Continued*)

**Table 7.** (Continued)

| | | | | | | | | | |
|---|---|---|---|---|---|---|---|---|---|
| Indeterminata | Indeterminata | Catkin Bract | Waterlogged | | | 1 | | | |
| Indeterminata | Indeterminata | Leaf | Waterlogged | 1 | | | 1 | 1 | |
| Indeterminata | Indeterminata | Bud | Waterlogged | | | 1 | | | |
| Indeterminata | Indeterminata | Mycorrhiza | Charred | | | | 1 | | |
| Charcoal | Indeterminata | Charcoal | Charred | * | ** | * | ** | ** | ** |
| Wood | Indeterminata | Wood | Waterlogged | | ** | * | ** | * | ** |
| Insect Exoskeleton | Indeterminata | Insect | Waterlogged | | × | × | × | × | × |
| Earthworm Cocoon | *Lumbricus sp.* | Cocoon | Waterlogged | × | | | | | × |
| Moss Animal | Bryozoa | Statoblast | Waterlogged | | | 1 | | | |

Sample numbers same as for soil chemical analyses. Amount of charcoal and wood was estimated using a relative scale, where

* signifies only a few pieces

** common occurrence and

*** large numbers. Presence of insect exoskeletons and earthworm cocoons were noted (×), not quantified.

UP = Upper peat, CL = white cultural layer, LP = Lower peat.

raspberry, hazelnut, crab apple and common nettle gives some indications of what was locally eaten or used to add flavours to the food.

Even though the preservation was, in general, good, it is evident that the plant material has been exposed to various transforming and corroding agents. For example, biological activity, such as burrowing of earthworms, is observable through earthworm cocoons (Fig 16) found both at the dryer and more highly elevated parts of the site (transition between upper peat and white cultural layer in trench 201) and in the wettest and deepest lying layer of zone 3 (lower peat in trench 209). The presence of earthworm cocoons also show that some of the content in the different trenches could be partly mixed and common nettle, hazel, raspberry and alder currently grows at the site and some of these remains could potentially be more recent intrusions. However, hazel occurs in large numbers only in deep-lying layers (Table 7), which suggests, despite earthworm presence, that these, are of Mesolithic origin, as also the earlier radiocarbon dates of hazelnuts from the site show (Fig 13).

Corrosion damage was observed on both water pepper seeds (Fig 17A) and hazelnut shells (Fig 17B) from trench 209. Currently, it is not known when this corrosion occurred, but it is conceivable that it coincides with the bone deteriorating processes.

Since plant remains are in general more resilient to soil acidification, compared to bones [68], the low pH in trench 209 has not dissolved the palaeobotanical record. However, further lowered pH might change this, especially if also combined with increasing admittance of oxygen into the soil by e.g. burrowing earthworms, further drainage or longer periods of lower precipitation.

## Discussion

By implementing a multiproxy approach we have analysed different aspects of archaeo-organic preservation and soil preservative qualities, which is also connected to the zooarchaeological analyses from the site [1]. Both our bone histological analyses and our analyses on collagen preservation have been conducted as comparative studies, where the results from the bone analyses from the 2019 excavation campaign have been compared to analyses on bones from

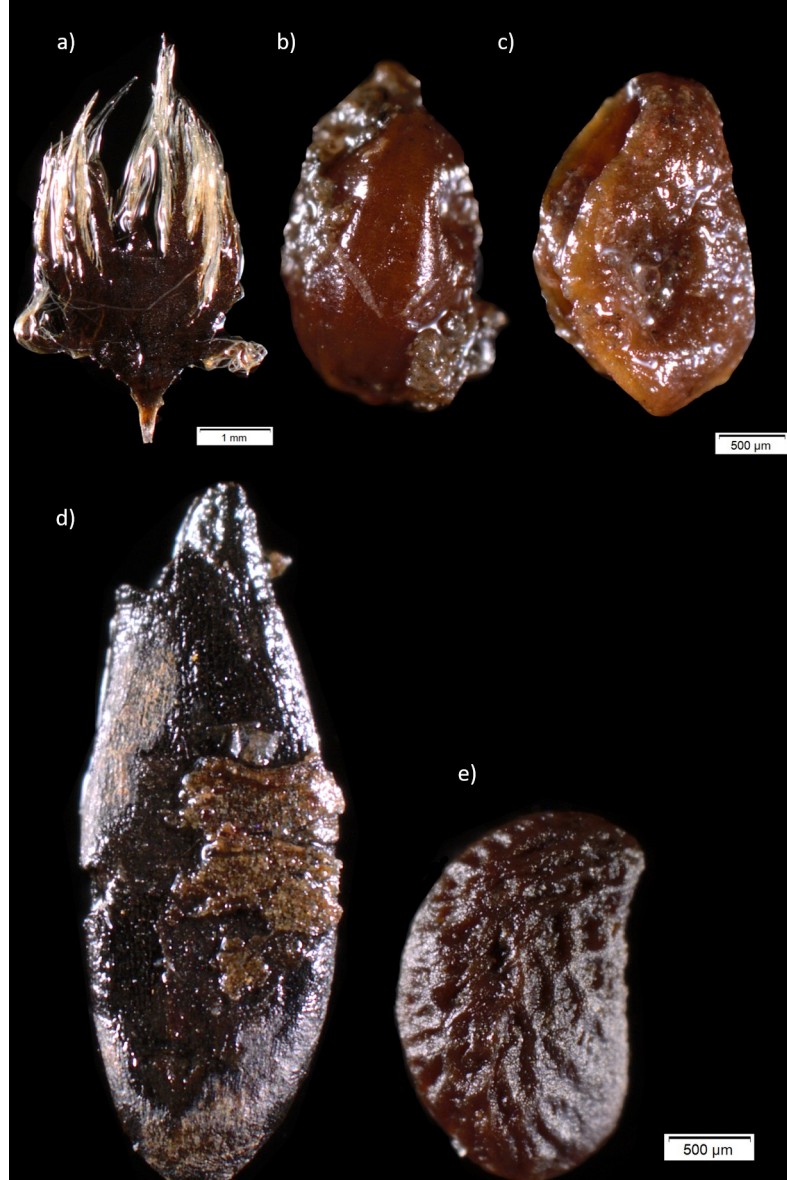

**Fig 15. A selection of archaeobotanical plant remains recovered from zone 3.** a) Aspen catkin bract; b) Pond Weed seed; c) White Water-lily seed; d) Crab Apple seed; e) Raspberry seed. Photo by Santeri Vanhanen for this publication.

the 1940s and the 1970s excavation campaigns. Because no soil samples were saved from the previous excavations at the site and no previous palaeobotanical analyses of plant macrofossils have been performed, the analyses on the 2019 material cannot be related to earlier conditions and may only serve as indicators of the current state of the soil and plant remains.

The decomposition of peat at Ageröd I:HC varies in state and intensity, as determined by the soil chemical properties (Tables 3 and 4), even within a small area and between adjacent layers through the peat sections. pH levels in the soil samples are of large importance for organic preservation, where a low pH is a key factor for intensified bone corrosion [69]. In pH levels below 6.5 the calcium phosphate in the bone hydroxyapatite begins to dissolve, this process increases significantly in pH below 6.0 and in pH below 4.5 bone material will rapidly

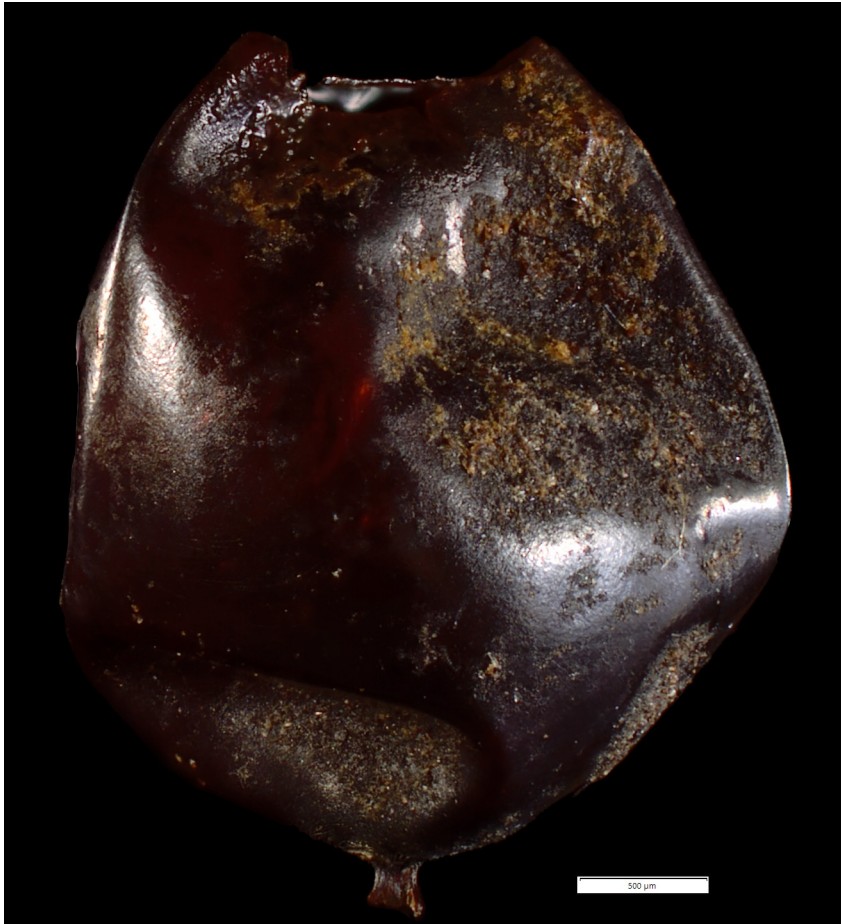

**Fig 16. Earthworm cocoon from the bottom of trench 209, sample 188.** Photo by Santeri Vanhanen for this publication.

deteriorate [70, 71]. This is well reflected in the soil samples from Ageröd where no bone remains were recovered from trench 259 and 209, both of which had very low pH (4.2 and 5.0). Assessments of the bone deposition pattern of the site, related to rectified GIS-data,

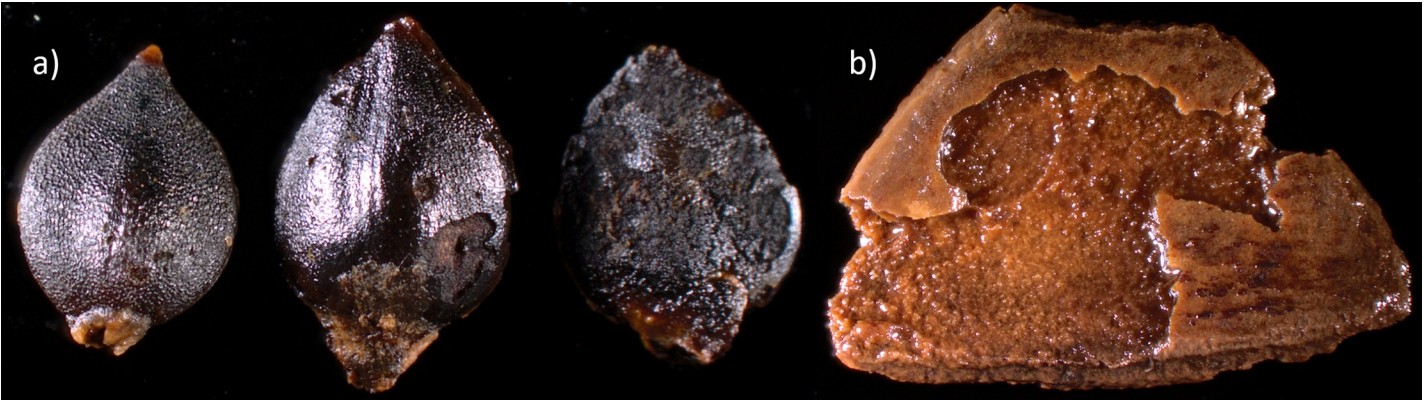

**Fig 17.** a) Water pepper seeds and b) hazelnut shell found in trench 209 with superficial corrosion damage. Photo: Santeri Vanhanen. Image originally created for [1] and freely available from PLOS ONE through CC by 4.0 licence.

enabled the positioning of the 2019 test trenches in the direct vicinity of where the best preserved and most numerous bone remains had been recovered on the previous excavations of the site [1], which suggest that it is unlikely that there would have been no bone remains originally deposited in the 2019 trenches.

As for the shifting redox situation and fluctuating groundwater table, the loss of manganese and the high MS quota (dominance of $Fe^{II}$) in the upper levels of trench 205 suggests a dynamic redox situation. Furthermore, the lowest sampled level of trench 205 appears to have lost its buffering qualities, as indicated by the low MSQ, caused by high MS values, low calcium levels and low levels of organic matter. This suggests that while pH is still favourable for bone preservation, the soil has no buffering qualities left and further acidification is likely to lower the pH drastically.

Low MSQ values in combination with high sulphur and iron content are demonstrated for trench 209. Because the lone sample (188) from this trench was taken in connection to the groundwater level, while the lowermost sample from trench 205 was almost 0.2 meters from the groundwater level, and displaying an even lower MSQ with only slightly lower sulphur and iron content, it implies that the much lower pH in trench 209 compared to 205 is connected to the groundwater. It is interesting to note that trench 209 had no unburnt bones, which is also true for the lower parts of trench 205 [1]; however, no soil samples were collected from the lowest sections of trench 205 and, consequently, its soil chemical properties could not be investigated. Furthermore, although the pH levels and organic content is still generally high in trench 205 and 209 (apart from the lowermost sample in 205, sample 187, where the organic content is low), the loss of manganese in both zone 3 trenches indicates that groundwater levels have varied considerably [62, 72, 73]. These fluctuations are also observable in the bone material, as the bone surface of all the bones recovered from trench 205 showed extensive surface etching (Fig 18) and high weathering degrees [1], but no biodegradation in the internal bone structure; which further suggests that the groundwater is acidic and is dissolving the bones from the outside. Since the bones in trench 205 have likely only been covered with acidic groundwater during high water level stands, the acidic water has 'only' etched and damaged the outer (0.5–1 cm) surface of the bones, whereby larger bones were still recoverable in the

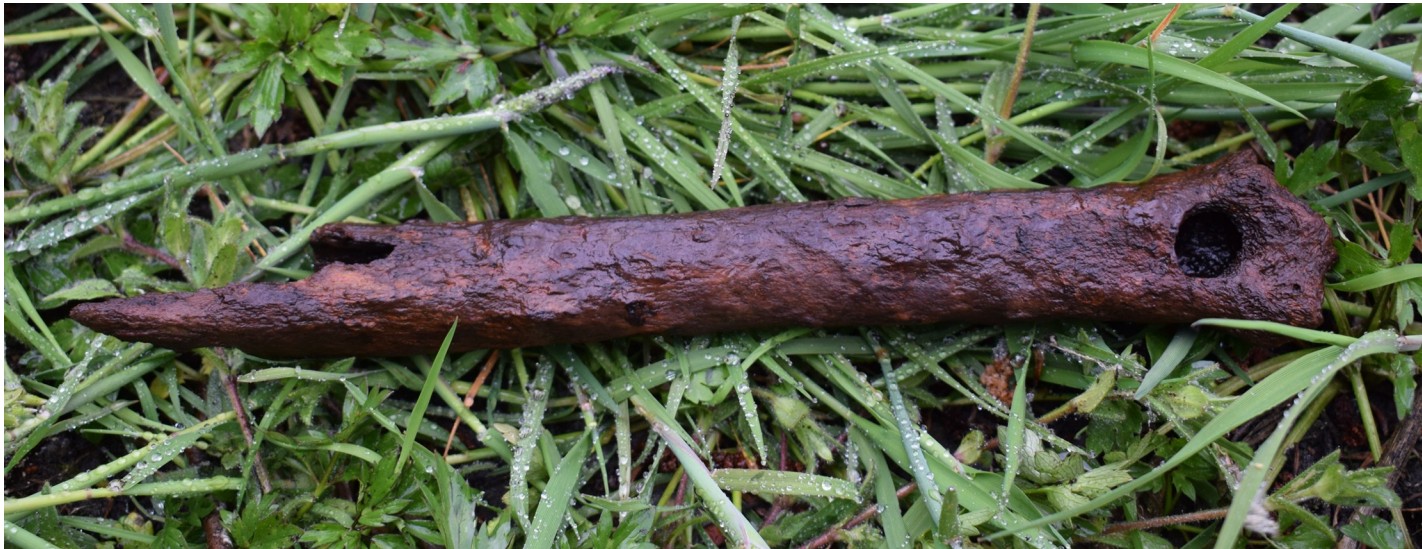

**Fig 18. Red deer tibia from trench 205 showing surface etching.** The picture was taken minutes after it was recovered and has here been gently washed to clean it in preparation for the picture. Photo by Mathilda Kjällquist for this publication.

trench and had not yet been completely dissolved, as the bones had been in trench 209, with low pH levels. However, the upper soil in trench 205 has now lost its buffering capacity, which suggests that if a new high water level occurs, the soil in areas corresponding to trench 205 might turn permanently acidic.

The general high iron content in both trench 205 and 209 in zone 3 differs from the other areas of the site. This contribution could be from the underlying rock, however; given that we have found oxidated pyrite in the bones it is more likely that the fluctuating groundwater in zone 3 has dispersed iron from the now oxidated pyrite in the bones into the surrounding soil. This process is also much more pronounced in 209, as illustrated by its much higher content of both iron and sulphur, which are both by-products when pyrite oxidates. Because sulphur has a negative effect on the pH level, the low pH in trench 209 is likely a function of the high levels of sulphur in that trench. Regarding the still high pH levels in trench 205 and the observations made from the other elements in the trench, it appears as if trench 205 is at a tipping point related to the buffer systems and a rapid change towards the situation in trench 209 may be well on the way.

The diagenetic alterations observed in histological thin-sections in part represent different burial conditions across the site. The bioeroded sample A6 excavated in the 1940s experienced a different burial environment to that of the other samples. This was found in the white cultural layer in zone 2, and the intense bioerosion is evidence of an oxic and free-draining environment with a neutral pH. This sample also seems to have experienced some changes in burial conditions, as the content of the destructive foci (the MFDs) at some point have been dissolved and leached out, leaving only the hypermineralized rims. This suggests acidic conditions. Some of the other samples have also experienced oxic conditions for a shorter period, as evidenced by limited bioerosion. The bacterial decay has at some point been halted by the establishment of anoxic conditions, most likely ensured by waterlogged conditions within the peat environment. This has remained fairly stable in at least parts of the site until the 1940s and 1970s, as intact pyrite is found within three of the six samples from these excavations.

Five of the eight histologically analysed bone samples from 2019 have experienced a detrimental change in burial conditions during their last years in the ground, as evidenced by the presence of oxidized pyrite grains. Four samples were analysed from different depths in one square (201), and of these, only the sample from the deepest peat layer was found to contain recognizable, but oxidized pyrite grains. The fact that the other three did not contain any pyrites may suggest different initial burial conditions in the layers higher up. However, it may also mean that the samples with no identifiable pyrite grains originally did contain the mineral but have earlier experienced fluctuating water conditions with wetting and drying cycles causing alternating oxic and anoxic periods, which might have completely dissolved the oxidized pyrite and allowed the released sulphuric acid to seep further down until reaching the groundwater below. Because zone 2 is located further up the slope from the former lake, this would then have happened earlier in the peat decomposing process, i.e. in closer connection to when the bog was initially drained in the early 20th century. However, all other analysed samples from 2019, both from trench 205, which is located further out into the bog, and from 217, where the bones derive from lower levels than in 201, also contain oxidized pyrite. Similarly, while only intact (non-oxidized) pyrite was recovered from the 1940s and the 1970s, in all cases they were found in the wet conditions of zone 3, or the lowermost levels of zone 2. Consequently, since no pyrite grains were recovered at higher stratigraphic levels in zone 2 it is also plausible to suggest a different (non-wet) original burial environment in the higher levels in zone 2 and zone 1, which would not have allowed pyrite to form at all.

The pyrite has gone from intact in the 1940s and 1970s to oxidized in 2019, which is important for our understanding of the drastic decrease in bone preservation, with severe etching

and cracking of the bones in zone 3 and significantly increased weathering levels in all areas of the 2019 excavation, compared to the bones recovered from previous excavation campaigns at the site [1]. When pyrite grains oxidize, they turn into iron-hydroxides at the same time as sulphuric acid is produced [52, 64]. The released sulphuric acid will etch and damage the bone, and cause pH levels to drop in the area. In trench 209, which is the wettest trench investigated in the 2019 excavation campaign, no bones were recovered. Furthermore, the levels of both sulphur and iron were most elevated in the soil sample from this trench and pH was low (5.0) which would have caused the calcium phosphate in the bones to rapidly dissolve. This indicates that the fluctuating groundwater levels (as observed by the depletion of manganese in zone 3) have led to dissolution and reprecipitation of minerals, and complete transformation of pyrite grains. Because trench 205 is also located in zone 3 it would also explain the severe dissolution, etching, staining and cracking observed in the still preserved bone samples from this trench and the complete absence of bones in the lowermost (waterlogged) part of the trench.

## Conclusion

As we have demonstrated with the histological analyses, the bones from both the 1940s and the 1970s excavations were affected by deterioration and in some cases also with severe destruction, suggesting that some changes to the burial environment were starting to have an effect already back then. The general trend, however, as also observed with the zooarchaeological analyses of the entire bone material from all excavation campaigns at the site [1], suggests that the bone material is deteriorating at an accelerating speed with significantly worse preservation of the bones recovered in 2019. The increase in bone degradation is most severe in zone 3, which is likely related to the oxidation of pyrite (which, as shown through the recovery of only intact pyrite grains within the bones from the old excavations while all of the detected pyrite grains in the 2019 bones had oxidized, did not start until after the 1970s) and the release of sulphuric acid which lowers the pH level locally. Because zone 3 is also experiencing fluctuations in groundwater levels (as observed by the depletion of manganese) it appears as if these fluctuations are increasing the rate of pyrite oxidation (observed by high levels of sulphur from the released sulphuric acid and iron from the released iron oxyhydroxide), which, due to the acidity of sulphuric acid, is lowering the pH level which might be incorporated into the groundwater during high water levels. The fluctuations causing cycles of wetting and drying, as well as dissolution and reprecipitation of minerals within the bone, are likely to further exacerbate the physical destruction of the bone at both micro- and macroscale (which, if considering that extreme weather situations, with both heavier rainfalls and periods of droughts, are estimated to further increase in the future [74, 75], is cause for concern). In areas where it is still wet, this has led to the destruction of the bone remains, though further investigations are needed to understand if the groundwater has turned acidic and dissolved the osseous remains throughout the bog or if these effects are limited to the vicinity of the old excavations. In slightly more elevated areas, where the bones are only in contact with the groundwater during high water stands (e.g. following extended periods of heavy rainfall), it has left the bones severely etched and corroded, but with larger bones still preserved.

The accelerated bone deterioration is, however, not limited to zone 3. This is clearly illustrated with the collagen preservation analyses, where only three of six bones from the 2019 excavation yielded any collagen (and on one of these occasions the collagen was likely contaminated) while all 12 analysed bones from the 1940s and the 1970s excavations had preserved collagen. Considering how much of modern archaeology is based on molecular level analyses, e.g. isotope analyses, aDNA and protein analyses [76–82], or, indeed, radiocarbon dating of

bone remains; and the rapid development of new scientific methods to further expand these fields, these results are not promising for the future (see Fig 14C).

It is, however, important to remember that all is not yet lost. Indeed, we have evidence of highly accelerated organic destruction at Ageröd even though the site has not been subjected to heavier exploitation and encroachment compared to other archaeological wetland sites (more the opposite, see discussions in [1]). Furthermore, Ageröd is far from the only archaeological site showing a pattern of accelerated organic destruction [12, 16, 20–22]. Nevertheless, the results presented here show that there is still an opportunity to somewhat remedy the situation. As we have shown, the archaeobotanical remains are still preserved in the wet contexts at Ageröd, even if the bones, which are more sensitive to increased acidity, have deteriorated here. Bones are still preserved in the dry contexts, and even if their preservation has declined significantly we are still able to extract collagen in the most well-protected areas of the site, i.e. in the deepest archaeological layers under the soil bank in zone 2, suggesting that it is likely still possible to also obtain e.g. aDNA if appropriate bones can be found in this area. Thereby, the opportunity lies in response to these results.

One defining question for the future of our ancient organic heritage is if it is possible to adapt our cultural heritage legislation designed to protect the buried archaeo-organic remains, e.g. can the Valletta Convention [83] be adapted to fully consider the implications of a stringent application of a 'preservation in situ approach', when we now know more about what is happening to this hidden cultural historical record? It would be possible to do excavations now, while there is still material and information to obtain and we still have the opportunity to develop and implement 'preservation in situ' methods for sites that cannot, for various reasons, be excavated in the present. However, saving the archaeo-organic remains is not easily done and although it is possible to understand the circumstances behind the degradation, the situation is difficult to change. The reason for organic preservation at Ageröd was, in the first place, made possible by anaerobic wet conditions protecting the archaeo-organic remains in the soil. However, if the area once again should become wet, the pyrite has oxidized and will release sulphur into the water, which will lower the pH even more, e.g. as observed at Star Carr [12], causing accelerated organic destruction. Hence, leaving the archaeological cultural remains from Ageröd I in dry conditions will ultimately destroy the organic remains, but submerging the site into acidic water will destroy the bone remains even faster. Thus, to enable future organic preservation, both at Ageröd and at similar sites elsewhere, pH levels in both soil and water must become, and continue to stay, neutral when groundwater levels are raised and actions to recreate anaerobic burial conditions are initiated.

Even when soil acidity is not part of the main equation, deterioration is still detectable. In trench 209, which had the best preservation of plant remains, the results showed that the remains are beginning to dissolve, as indicated by initial corrosion on some of the seeds (Fig 17). Indeed, when a layer becomes aerobic, different organisms can survive and consume plant tissues. This suggests that once the oxidation process has started it is a race against time. Consequently, while it is still not too late to recover information from the site, the window of opportunity is rapidly closing.

We do not have the luxury of waiting for a more convenient time. The organic remains are rapidly deteriorating, at least in Northern Europe and around the northern hemisphere, but likely all over the world. Once this record is gone there is no turning back. We cannot, in twenty years, come back and say 'now we will deal with this' because then it will be too late. Indeed, in many cases and areas of the world it is likely too late to be proactive; so let us now be reactive instead of regretful! The time to act is now, and if we wait we will be in a situation where our organic long-term record of climate, environment and human culture will be lost forever. A loss that is absolute and irretrievable. We only have one opportunity to save and protect the future of our past, and it is now.

## Acknowledgments

All necessary permits were obtained for the described study, which complied with all relevant regulations of the County Administrative Board of Scania, Sweden. We would like to thank the reviewers for their insightful comments. We acknowledge and thank the curators at the Historical Museum at Lund University for permission to analyse and sample bone remains in their care. Lastly, we are grateful for the assistance of our collaborators in the Stone Age for the Future Network and everybody else who volunteered and assisted during our excavation of the site.

## Author Contributions

**Conceptualization:** Adam Boethius.

**Data curation:** Adam Boethius, Mathilda Kjällquist, Ola Magnell.

**Formal analysis:** Adam Boethius, Hege Hollund, Johan Linderholm, Santeri Vanhanen.

**Funding acquisition:** Adam Boethius, Jan Apel.

**Investigation:** Adam Boethius, Mathilda Kjällquist, Ola Magnell.

**Methodology:** Adam Boethius.

**Project administration:** Adam Boethius.

**Writing – original draft:** Adam Boethius, Hege Hollund, Johan Linderholm, Santeri Vanhanen.

**Writing – review & editing:** Adam Boethius, Hege Hollund, Mathilda Kjällquist, Ola Magnell, Jan Apel.

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
