## [Decision Letter · Decision Letter 0]

17 Aug 2020

PONE-D-20-20896

Quantifying archaeo-organic degradation – a multiproxy approach to understand the accelerated deterioration of the ancient organic cultural heritage at the Swedish Mesolithic site Ageröd

PLOS ONE

Dear Dr. Boethius,

Thank you for submitting your manuscript to PLOS ONE. After careful consideration, we feel that it has merit but does not fully meet PLOS ONE’s publication criteria as it currently stands. Therefore, we invite you to submit a revised version of the manuscript that addresses the points raised during the review process.

Please address all comments before re-submission including restructuring of the 'materials and methods' chapter.

We look forward to receiving your revised manuscript.

Kind regards,

Peter F. Biehl, PhD

Academic Editor

PLOS ONE

Additional Editor Comments:

Please address all comments before re-submission including restructuring of the 'materials and methods' chapter.

Journal Requirements:

2. We note that Figures in your submission contain copyrighted images. All PLOS content is published under the Creative Commons Attribution License (CC BY 4.0), which means that the manuscript, images, and Supporting Information files will be freely available online, and any third party is permitted to access, download, copy, distribute, and use these materials in any way, even commercially, with proper attribution. For more information, see our copyright guidelines: http://journals.plos.org/plosone/s/licenses-and-copyright.

2.1.         You may seek permission from the original copyright holder of Figures to publish the content specifically under the CC BY 4.0 license.

2.2.    If you are unable to obtain permission from the original copyright holder to publish these figures under the CC BY 4.0 license or if the copyright holder’s requirements are incompatible with the CC BY 4.0 license, please either i) remove the figure or ii) supply a replacement figure that complies with the CC BY 4.0 license. Please check copyright information on all replacement figures and update the figure caption with source information. If applicable, please specify in the figure caption text when a figure is similar but not identical to the original image and is therefore for illustrative purposes only.

3. We noted in your submission details that a portion of your manuscript may have been presented or published elsewhere.

"Yes, this paper is step two in our study of the archaeo-organic deterioration at the Mesolithic site Ageröd. This first paper titled "Human encroachment, climate change and the loss of our archaeological organic cultural heritage: Accelerated bone deterioration at Ageröd, a revisited Scandinavian Mesolithic key-site in despair" was recently accepted for publication in PlosOne. Because these two papers are interlinked, but we still want them to be able to stand on their own some of the figures in both these papers are similar or the same (Figs 1-3, 15).  The other paper about the Ageröd bone deterioration is attached as a related work. "

Please clarify whether this publication was peer-reviewed and formally published. If this work was previously peer-reviewed and published, in the cover letter please provide the reason that this work does not constitute dual publication and should be included in the current manuscript.

Reviewers' comments:

Reviewer's Responses to Questions

**Comments to the Author**

1. Is the manuscript technically sound, and do the data support the conclusions?

Reviewer #1: Yes

Reviewer #2: Yes

2. Has the statistical analysis been performed appropriately and rigorously? 

Reviewer #1: Yes

Reviewer #2: N/A

3. Have the authors made all data underlying the findings in their manuscript fully available?

Reviewer #1: Yes

Reviewer #2: Yes

4. Is the manuscript presented in an intelligible fashion and written in standard English?

Reviewer #1: Yes

Reviewer #2: Yes

5. Review Comments to the Author

Reviewer #1: General comments: At least one author seems to have real trouble distinguishing between singular and plural. A lot of references to own publications rather than others which might be more relevant. All references to Boethius et al. (accepted) should be changed to Boethius et al. in print (unless that is a specific PLOS ONE way of referring to accepted manuscripts).

-> means 'change to'.

Most are minor language or grammatic corrections.

Swedish 'kulturlager' translates into 'archaeological layer' or 'archaeological deposit', NOT 'cultural layer'!

l.15: sites,

l.16: preservation,

l.23: have -> has

l.24: ..pHvalues in the still wet areas of the site have.. (delete two commas)

l. 26 oxidizes

l.31: ...within a decade through...

l.37, l.50, l.60, l.72, l.78, l.89, l.102, l.110, l.139, l. 373, l.386, l453, l.461, l.494, l.547, l. 563, l.585,l.592, etc.: accepted -> in print (systematic change of all following references to this text)

l.42: recovered on -> recovered at

l.45: perquisites -> prerequisites

l.63: made,

l.53: are -> is

l.65: was -> were

l.83: site,

l.90: available upon reques to the museum -> available at the museum upon request

l.93: decision -> decisions

ll.96-133: curious lack of any references to geoarchaeology and geochemical analyses outside Swedish borders and why these methods are preferred over others

l.98: samples -> sample

l.99: trench 201 and 205-> trenches 201 and 205

l.101: from trench -> from trenches

l.105: delete cultural

l.107: cultural -> archaeological

l.108: layers,

l.113: what -> which

l.114: remains,

Table 1, Cit-P: delete double reference to Arrhenius, 1934

l.169: from 1970s -> from the 1970s

l.176: following -> ensuing (because following is used in next line in same sentence)

ll.188-189: with- wash over -> with wash-over

l.189: Tolar et al., 2009

l.193: fraction -> fractions; was -> were; partly -> only partly/ only in part

l.203: who -> which

l.206: Result -> Results (again in this chapter, a curious lack of any references to geoarchaeology and geochemical analyses outside Swedish borders and why the used methods are preferred over others used in the UK, DK, NL, NO...)

l.216: delete cultural

l.218: delete cultural

l.244: iron,

l.251: is -> being

l.255: trench -> trenches

l.256: which is -> something which is

l.257: slope,

l.258: reducing -> degrading (otherwise doesn't make sense when it is stated that the soil has completely oxidised - can't be reducing and oxidised at the same time); condition -> conditions

l.268: it -> there; active process -> active degradation process

l.275: relates -> relate

l.290: sample -> samples; who -> which

l.291: axle -> axis

l.303: lengthening (and -> lenthening, and; has -> have

l.307: delete cultural

Table 5: Wildboar -> Wild boar (3 occurences); good that Alces alces is stated as Elk/Moose cf. l. 363

l.336: (F),

l.354: like a budding -> like budding

l.360: appear -> appears

l.361: delete cultural

l.362: layer,

l.370: analysed,

l.379: redox (good, should be used consistently in text - please change all uses of red-ox to redox)

l.384: show -> shows

Table 6: Wildboar -> Wild boar; Elk -> Elk/Moose (cf Table 5 and l.363 (5 occurences))

l.395: suggest -> suggests

Fig 11 caption, ll. 399-402: please explain the difference between WL and CL??? The term 'archaeological cultural layer' is incorrect, delete cultural

l.403: campaigns,

l.436: good; -> good,

l.439: cultural -> archaeological

l.465: done -> performed

l.477: reducing??? All following text indicates oxidising conditions. Please check!

ll. 477, 481, 489: Furthermore (used thrice, consider variation)

l.495: suggest -> suggests; are -> is

l.497: water level high stands -> high water level stands

l.500: occurs,

l.503: have -> has

l.510: as -> when; oxidate -> oxidates

l.517: delete cultural

l.518: 2,

l.523: conditions,

l.527: its -> their

l.530: differential -> different; condition -> conditions

l.560: excavation -> excavations

l.563: suggest -> suggests

l.573: wheater -> weather

l.583: excavation -> excavations

l.593: add reference: McGovern, 2018

l.596: who -> which

l.597: have -> has

l. 619: shown -> indicated

Reviewer #2: This paper presents an interesting and comprehensive analysis, which I think would be of interest to readers of PLOS one. The study is of great importance and highlights serious concerns about the future of wetland heritage. I enjoyed reading about such a fascinating site, and it is so unfortunate that its future is uncertain.

General points

Style and organization: I think that some improvement could be made in terms of the writing; please note that I have been picky in this regard because it was generally a rather pleasant read. In places however, the text is overly wordy and repetitive. I have pointed out some examples, but I do feel that the manuscript could do with another read over and some text removed to improve its readability. This is a particular problem in the “Discussion” section, where I feel that phrases such as “furthermore” and “in this context” have been added too often when sentences don’t actually link, and this has made it quite confusing to read. There is also some inconsistency in the writing style, particularly use of tenses (e.g. “we did” is used more in “collagen preservation” than other sections - this section doesn’t seem to match the rest of the manuscript). I don’t personally have a preference for passive tense as long as it is consistent but please refer to journal style. A lot of the background information included in the “results” should really be in the “Discussion” section, and indeed in many places information is repeated in both. Please check for repetition and that information is in the appropriate place.

Introduction: In my opinion this is too brief. I would like to see more information of the other recent study at the site and how it relates to this one. I think the structure would be better if the site description was integrated here so that we already know its importance and history of excavation – it would make it easier to understand the context of this present study. Some of the information in the “site description” section would be better in “Materials and Methods”. I would also prefer a brief discussion of the analytical techniques used in the introductory section as they are an important part of the paper.

On occasion I think there has been over interpretation of results and would like to see this resolved before publication.

Finally, I think the title should be reconsidered. Archaeo-organic is a rather niche term that might not be very searchable, and I think the site name should be in the main title. In fact, I suggest you consider using your subtitle as the main title of the article: “A multi-proxy approach to understanding the accelerated deterioration of the ancient organic cultural heritage at the Swedish Mesolithic site Ageröd”

Specific points

Line 23: “have already lost” should be “has already lost”

Line 26: “oxidize” should be “oxidizes”

Line 34: Please be consistent with the name of the site. Here you call it Ageröd I:HC but earlier it is just Ageröd, or Ageröd I. Are these different parts of the site? If so, please define what the I and/or HC means.

Line 38: there is a word missing here, perhaps “..degradation and revealed that in some areas….”

Line 39: What is meant by “hidden” here?

Line 45: “Prerequisites”?

Line 65: “which was used” should be “which were used”

Line 76: Do you really mean to say “arbitrarily”?

Lines 92 and 96: Were there specific sampling criteria? Were the bone and soil samples selected at random?

Line 98: “samples” should read “sample”. This description of samples is also quite confusing, perhaps reference to one of the tables would help.

Lines 113-115: here is an example of an overly wordy sentence (see general comment). I also do not know what is meant by “A variety of spectroscopic techniques of X-ray fluorescence….” As I don’t believe wet chemical, pH or MS are spectroscopic techniques?

Table 1: I really like this, it’s very helpful. However, for Cit-P and Cit-POI you have explained the units of measurements in the “Method” column, whereas for previous methods it is in the “description” column.

Lines 126-129: An example of repetition, this information is all in the Table. Can information for XRF be put into the table?

Line 135: I appreciate that that when dealing with so many different sample types it is confusing to explain, but in places it is not clear where samples are from or which are being discussed. Better reference to the tables could help with this. There is also some repetition which doesn’t help, you have already mentioned the selection of six samples from the previous excavations, I assume these are the same six – why not just mention them here for the first time?

Line 141: The description of features would really be helped with some images. I know these do appear in Figures 7-9 – might it be better to discuss them in this level of detail in the results section so that the figures are there? I also don’t think you have adequately explained the difference between OHI and GHI. Putting them together in Table 2 has made this even more confusing.

Line 168: It is unclear whether these are the same bones used for histology.

Line 188: It would be clearer to say that 0.1 litre per sample was analysed (if this is correct!)

Line 193: “whereas 1-0.4…. these samples” is unclear. What does (ca1/10) mean? Why does the high organic content matter?

Line 196: *, ** and *** - what do they mean? Perhaps they refer to a figure; if so, the reference is missing.

Line 206: “Result” should read “Results”

Line 208-209: An example of repetition / superfluous wording. This sentence is not necessary.

Line 213: The meaning of “the potential corrosiveness of the sediment-bone relationship” is unclear. The large span does not necessarily indicate that the sediment is corrosive, rather the lowest pH value does.

Table 4: For readers who are unfamiliar with XRF data, I think you should remind us what the units are here.

Line 228: What does “different prerequisites” mean here? Very ambiguous, do you mean different soil chemistries?

Figure 4: I think this is a nice presentation of data but somewhat confusing. It could be improved by making it clearer that the colour refers to the trench number and putting the range of values onto each graph rather than in the legend. The legend “depth variation correlated for pH and different elements” is ambiguous – I think something like “Variations in soil chemistry as a function of pH and depth” is a more accurate description. I also do not agree with the use of the term “element” here as organic matter is not an element, and a lot of the description in the text underneath the legend should be in the legend itself, and does not need repeating in the text.

Line 239: “shows pH as a function of…” is not true. The use of “as a function of” throughout this paragraph needs to be reconsidered. The figure shows pH as a function of depth, the magnitude of the soil chemistry parameters are plotted independently of these.

Line 264-265: It is unclear what the connection is between S content and “an initial and rapid decay of the OM”.

Line 271 (para beginning): I feel that the PCA has not been adequately described or justified. I actually do not see what it adds to the analysis and suggest either clarifying this or removing it entirely – you seem to conclude that pyrite causes differences in XRF data but that is obvious anyway. Many of the other conclusions / observations can be discerned from Figure 4 anyway, and to someone unfamiliar with PCA, there is no obvious separation of data points in either Figure 5 or 6. If it is left in, it needs far more explanation (e.g. p2t2 to p3t3? What does negative orientation mean? How does it relate to organic deterioration?)

Line 273: The use of phosphate content for “assessing general cultural impact” needs explaining.

Table 5: The weathering analysis needs explanation (i.e. which is worse, 1 or 5, what to the numbers in brackets mean?)

Line 311: A figure showing birefringence would be good (but I know it’s hard to show in a still).

Line 320-322: An example of an overly repetitive sentence. There are a few like this throughout and I do feel that it would benefit from a re-read.

Line 369: Is there a reason more bones weren’t studied?

Line 372: “deviant pattern… limited” very unclear what is meant by this, and how this relates to the weathering analysis.

Table 6 contains a lot of information and I wonder if all of the columns are really necessary, particularly masses of sample and amount of collagen in mg as really it’s only the % that’s important to the interpretation.

Lines 389-393: Is all of this information really necessary?

Line 403 (para beginning): This is very repetitive – most of this is in the figure legend as well.

Figure 12: I do not agree that this data is enough to establish a downwards trend and think that this has been over interpreted. There are not many data points for later excavations, and correlation is quite weak across only three time points. The prediction that the site will be lost by 2040 assumes deterioration is linear, and you can’t know that from this study. It can perhaps be presented as a hypothesis, but I have a problem with the way this has been presented here as fact.

Table 7: legend says 6 samples, it should be 7

Figure 13: Do you really mean “collation” - maybe “selection”?

Line 447: The link between the presence of corrosion damage and the material being of Mesolithic origin needs clarifying. Can corrosion not have been caused at any time period?

Line 466: “Indications” should be “indicators”

Line 458: In general, I find this discussion repetitive from the previous sections and the whole thing could do with editing, with some sentences shortened or deleted.

Line 468: The decomposition of peat as determined by what parameter – can you refer to a table? Do you maybe mean that the soil chemistry varies?

Line 473: I think you need to acknowledge that you may not have found bones in these trenches simply because there never were bones. I don’t think it takes from the study at all, because there clearly IS significant deterioration, but there are limitations on the data – your data set is (understandably) very small.

Line 481: Again I think there has been some over interpretation here. There is only one sample from trench 209, so can you really say it is “characteristic”? How do you know that the difference isn’t due to the groundwater level? I think perhaps this point has not been discussed very clearly.

Line 487: decimeters is quite an unusual term to use, and for the first time in this manuscript.

Line 499: You cannot know that bones have been dissolved – what evidence is there that they ever existed? The phrase “But remember that…” is a bit odd in the tone of the paper.

Line 506: “suggests that there is a contribution of iron to the soil” is superfluous. Please delete.

Line 516: “completely bioeroded” implies that it has disappeared!

Line 524: Are you sure that this means conditions have been stable until the 1940s/70s? Can pyrite not have formed at any time between deposition and excavation?

Line 565: I am not convinced from the discussion here that it has actually been proven that pyrite oxidation occurred so recently. Perhaps this needs communicating a little better?

Line 573: “wheater” should be “weather”

Line 575: Check reference

Line 576: “Complete destruction of the bone remains” – please reconsider this hypothesis!

Line 597: “have declined” should read “has declined”

Line 602 (para beginning): These are very important conclusions, and passionately communicated. May I suggest that you might want to highlight the fact that changing climates will make it more and more difficult to maintain stable conditions at sites like this? Do you have suggestions on what can be done – do you propose simply to excavate? I also suggest that you repeat some of this urgency and passion in the abstract of the manuscript, in case it is all that someone reads.

6. PLOS authors have the option to publish the peer review history of their article (what does this mean?). If published, this will include your full peer review and any attached files.

Reviewer #1: **Yes: **Vibeke Vandrup Martens

Reviewer #2: No

---

## [Author Response · Author response to Decision Letter 0]

7 Sep 2020

Please see attached file (end of built pdf) with colour coded replies to each comment made by the reviewers

---

## [Editor Report · Decision Letter 1]

10 Sep 2020

Quantifying archaeo-organic degradation – a multiproxy approach to understand the accelerated deterioration of the ancient organic cultural heritage at the Swedish Mesolithic site Ageröd

PONE-D-20-20896R1

Dear Dr. Boethius,

We’re pleased to inform you that your manuscript has been judged scientifically suitable for publication and will be formally accepted for publication once it meets all outstanding technical requirements.

Kind regards,

Peter F. Biehl, PhD

Academic Editor

PLOS ONE
---

## [Editor Report · Acceptance letter]

14 Sep 2020

PONE-D-20-20896R1 

Quantifying archaeo-organic degradation – a multiproxy approach to understand the accelerated deterioration of the ancient organic cultural heritage at the Swedish Mesolithic site Ageröd 

Dear Dr. Boethius:

I'm pleased to inform you that your manuscript has been deemed suitable for publication in PLOS ONE. Congratulations! Your manuscript is now with our production department. 

Kind regards, 

on behalf of

Dr. Peter F. Biehl 

Academic Editor

PLOS ONE